# PLOT: Prompt Learning with Optimal Transport for Vision-Language Models

**Guangyi Chen[†•], Weiran Yao[†], Xiangchen Song[†], Xinyue Li[◇], Yongming Rao[‡], Kun Zhang[†•]**

[†]Carnegie Mellon University, Pittsburgh PA, USA
[•]Mohamed bin Zayed University of Artificial Intelligence, Abu Dhabi, UAE
[‡]Tsinghua University, Beijing, China
[◇]New York University, Abu Dhabi, UAE

## Abstract

With the increasing attention to large vision-language models such as CLIP, there has been a significant amount of effort dedicated to building efficient prompts. Unlike conventional methods of only learning one single prompt, we propose to learn multiple comprehensive prompts to describe diverse characteristics of categories such as intrinsic attributes or extrinsic contexts. However, directly matching each prompt to the same visual feature is problematic, as it pushes the prompts to converge to one point. To solve this problem, we propose to apply optimal transport to match the vision and text modalities. Specifically, we first model images and the categories with visual and textual feature sets. Then, we apply a two-stage optimization strategy to learn the prompts. In the inner loop, we optimize the optimal transport distance to align visual features and prompts by the Sinkhorn algorithm, while in the outer loop, we learn the prompts by this distance from the supervised data. Extensive experiments are conducted on the few-shot recognition task and the improvement demonstrates the superiority of our method. The code is available at https://github.com/CHENGY12/PLOT.

## 1 Introduction

In the past few years, large-scale vision-language pre-trained (VLP) models, such as CLIP (Radford et al., 2021), ALIGN (Jia et al., 2021), and BLIP (Li et al., 2022) have achieved remarkable success in open-world visual concept learning. These methods have brought new light but also pose a new question: how to efficiently adapt the knowledge from pretraining to the downstream tasks since these models are typical of massive sizes which are not feasible for normal users to re-train.

One of the conventional paradigms of utilizing pretrained knowledge is "pre-training, fine-tuning", which fixes the architecture of the pre-trained neural network and tunes its parameters using task-specific objective functions. Beyond fine-tuning the parameters, many recent methods (Zhou et al., 2021b; 2022) introduce the concept of prompt learning from the field of NLP to the vision domain and achieve striking performance gain for the few-shot visual classification. They fix the model parameters and instead learn suitable prompts by turning a template sentence into a set of learnable vectors. Then, these prompts are learned by minimizing the distance between the visual features and prompt-based language features.

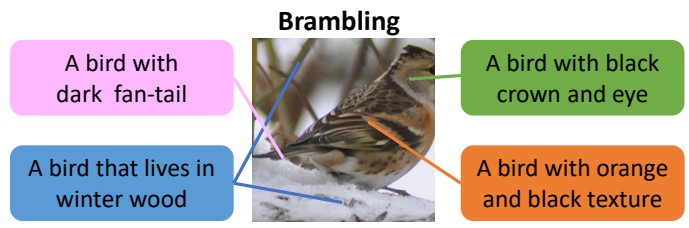

**Brambling**

A bird with dark fan-tail

A bird that lives in winter wood

A bird with black crown and eye

A bird with orange and black texture

Figure 1: The motivation that one category can be complementarily described in different views (An example of "Brambling").

Despite significant improvements over manual prompts, learning only a sentence is intuitively insufficient to represent a class. One class can be described by many intrinsic characteristics and even extrinsic context relations. Thus, for one object, we may have multiple prompt candidates which focus on different attributes. As shown in Figure 1, we can describe the class "Brambling" in

different views: such as the color of the wing, the color of the crown and eyes, the shape and color of the tail, and even the living environment information. It motivates us to learn multiple prompts to comprehensively represent the class and thus facilitate classification.

The most natural solution is to directly learn multiple prompts by respectively matching each prompt with the visual features. However, it is the same as matching the mean of prompt features and the visual features. This solution is problematic since all prompts are encouraged to be closer to one single point and thus tend to learn the same characteristics. It contradicts our purpose to learn comprehensive prompts. To solve this problem, we tested adding some constraints to push away the prompt from each other, but found that this solution still fails to learn representative and comprehensive prompts. This solution treats the visual representation as one single point, and such a unified view of visual features ignores the fact that different prompts may only focus on one or a subset of characteristics.

To address this problem, in this paper, we propose Prompt Learning with Optimal Transport (**PLOT**), which applies optimal transport (OT) to align the local visual features and multiple textual prompts. Optimal transport can calculate the distance between two distributions under the form of multiple sampling. In our prompt learning framework, we formulate local visual features and multiple prompts as the samplings of two discrete distributions and use OT to encourage fine-grained cross-modal matching. Specifically, to obtain the local visual features with different semantic clues, we extract all feature maps as the visual representation instead of the single global representation. Fortunately, we can easily obtain the visual feature maps from the visual encoder of CLIP by using all outputs of the multi-head self-attention layer (Rao et al., 2021). Then the problem comes down to how to calculate the distance between two feature sets.

We solve this problem by introducing the optimal transport theory (Villani, 2009) and formulate the feature sets as a discrete probability distribution where each feature has an equal probability value. Furthermore, to reduce the computational cost and avoid the extra model parameters, we learn the prompts with a two-stage optimization strategy. At the first stage in the inner loop, we fix both visual and text features and optimize the optimal transport problem by a fast Sinkhorn distances algorithm (Cuturi, 2013). Then, in the outer loop, we fix all parameters of optimal transport and back-propagate the gradient to learn the prompts with different characteristics. Compared with conventional distance (such as Euclidean distance of mean features), optimal transport can align different visual features for each local prompt, which is more robust to the visual misalignment and tolerates well feature shift (Rubner et al., 2000). It is because OT learns an adaptive transport plan to align features, which achieves fine-grained matching across two modalities. We conduct experiments on 11 datasets following the standard setting of CLIP (Radford et al., 2021) and CoOp (Zhou et al., 2021b) to evaluate our method. These experiments span the visual classification of generic objects, scenes, actions, fine-grained categories, and so on. The significant result improvement demonstrates that **PLOT** can effectively learn representative and comprehensive prompts.

## 2 RELATED WORK

**Optimal Transport** The Optimal Transport (Monge, 1781) is initially introduced to solve the problem of how to reduce the cost when moving several items simultaneously. Recently, OT theory has drawn wide attention in the machine learning and computer vision community by comparing distributions readily available to them under the form of feature sets (Peyre & Cuturi, 2019). Due to the brilliant property of distribution matching, OT has been applied in many theoretic and application tasks including generative models (Arjovsky et al., 2017; Salimans et al., 2018; Zhao et al., 2021a), structural matching (Chen et al., 2019; Xu et al., 2020; Zhao et al., 2021b; Xu et al., 2019) (e.g. sequence matching (Chen et al., 2019) and graph matching (Xu et al., 2019), and image matching (Zhang et al., 2020; Liu et al., 2021a; Zhao et al., 2021b)), and other distribution-based tasks (such as clustering (Laclau et al., 2017), distribution estimation (Boissard et al., 2015), and causal discovery (Tu et al., 2022)). In this paper, we use OT to align the features of vision and language modalities by learning an adaptive transport plan (Rubner et al., 2000).

**Vision-Language Pre-trained Models** Vision-Language Pre-trained (VLP) models aim to explore the semantic correspondence between the vision and language modalities through large-scale pre-training. Recently, VLP models have achieved an exciting performance improvement in few-shot visual recognition (Radford et al., 2021; Gao et al., 2021; Zhou et al., 2021b; 2022; Zhang et al., 2021b), which shows the great potential to promote open-world visual understanding with the help of language. In terms of objectives, VLP methods can be divided into reconstruction (Li et al.,

2019; Hong et al., 2021; Dou et al., 2021; Kim et al., 2021), contrastive matching (Radford et al., 2021; Jia et al., 2021; Jain et al., 2021), or the combination of both two (Li et al., 2021; Wang et al., 2021b; Kamath et al., 2021). Besides, recent progress in the field of VLP also benefits a lot from large-scale pair-wised datasets. For example, CLIP (Radford et al., 2021) applies 400 million image-text pairs for contrastive learning. Beyond recognition, these VLP models also show great potential for other downstream applications, such as dense prediction (Rao et al., 2021; Zhou et al., 2021a), image generation (Nichol et al., 2021; Ramesh et al., 2022; Patashnik et al., 2021), and action understanding (Wang et al., 2021a; Tevet et al., 2022).

**Prompt Learning** Prompt learning is introduced from the field of NLP to efficiently adapt the large language model to downstream tasks. Different from the conventional "pre-training, fine-tuning" paradigm which initializes the pre-trained model and tunes the parameters of the network using downstream task-specific objective functions, prompt learning applies textual prompt to reformulate the downstream tasks as the original pretrained task (Liu et al., 2021b; Petroni et al., 2019). By the prompt, the domain shift between pretrained task and the downstream application is reduced and thus the pretrained knowledge can be easier adapted to downstream tasks. The concept of prompt learning (Petroni et al., 2019; Radford et al., 2019; Poerner et al., 2019) begins from the success of GPT (Radford et al., 2019) series. Early prompt learning methods (such as Petroni *et al.* (Petroni et al., 2019) and Pörner *et al.* (Poerner et al., 2019)) always manually create templates based on human prior knowledge. Furthermore, some mining-based methods (Jiang et al., 2020) and gradient-based methods (Shin et al., 2020) are proposed to automatically search for appropriate templates. Beyond search in the discrete space, some methods (Li & Liang, 2021; Tsimpoukelli et al., 2021; Liu et al., 2021c) remove the constraint that the prompts are "words" and instead learn prompts in the continuous embedding space. Recently, CoOp (Zhou et al., 2021b) and its extended version (Zhou et al., 2022) introduce prompt learning into open-world visual understanding to adapt the knowledge from the large-scale visual-language pretrained models and achieve great performance improvement on the few-shot visual recognition. Compared with CoOp, our **PLOT** method further improves prompt learning by introducing the optimal transport distance to learn multiple local prompts and achieves fine-grained vision-language matching. PDL (Lu et al., 2022) is also motivated by the more diverse prompts, which assumes a parametric distribution of prompts and fits the parameters during training. Different from it, **PLOT** learns multiple prompts without parametric distribution.

## 3  APPROACH

In this section we first revisit the baseline CoOp (3.1), review the preliminaries of optimal transport (3.2), and then introduce our **PLOT** (3.3) to show how we learn multiple comprehensive prompts.

### 3.1  A REVISIT OF COOP

CoOp (Zhou et al., 2021b) is one of the pioneering methods to learn the prompts for using vision language pretrained knowledge (such as CLIP (Radford et al., 2021)) for downstream open-world visual recognition. Different from CLIP which manually designs the prompt templates, CoOp sets a part of context words in the template as continuous learnable parameters which can be learned from the few-shot data. Then the classification weights can be represented by the distance between the learned prompt and visual feature.

Specifically, given an image $x$, a visual feature $f = f(x)$ is obtained by the visual encoder $f$ of CLIP. Then, the textual prompt can be formulated as $t_k = \{\omega_1, \omega_2, \ldots, \omega_L, c_k\}$, where $c_k$ is the word embedding of the class name, $\omega = \{\omega_l|_{l=1}^L\}$ are learnable vectors where each vector has the same dimension as the original word embedding and L is the length of context words. With prompt $t_k$ as the input, the text encoder $g$ outputs the textual feature as $g_k = g(t_k)$. The final prediction probability is computed by the matching score as follows:

$$p(y = k|x) = \frac{\exp(\text{sim}(f, g_k)/\tau)}{\sum_{k'=1}^K \exp(\text{sim}(f, g_{k'})/\tau)}, \qquad (1)$$

where $\text{sim}(\cdot, \cdot)$ denotes a metric function such as cosine similarity, and $\tau$ stands for the temperature of Softmax. Then we can optimize the parameters of $\{vec_l|_{l=1}^L\}$ with the cross-entropy loss between the prediction and the labeled target.

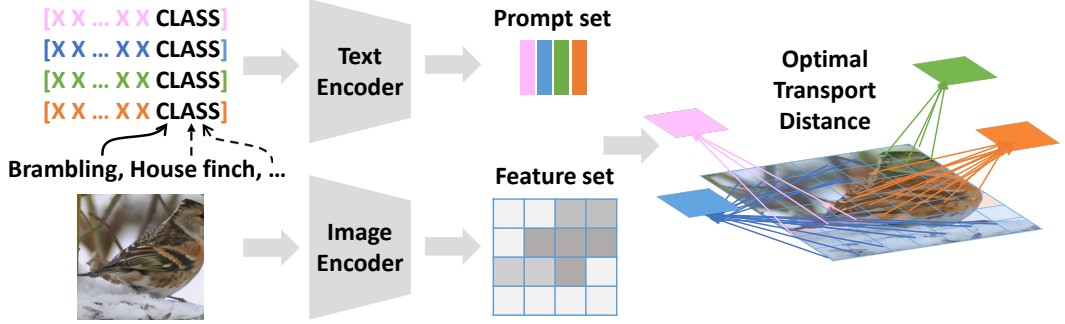

Figure 2: **The framework: PLOT** first describes each category with multiple prompts and obtains a set of prompt features by text encoder. The image is also encoded as a set of local features. Then the optimal transport is used as the metric between prompts and visual features.

## 3.2 OPTIMAL TRANSPORT

Optimal transport (OT) distance is a widely used metric for the comparison of distributions. Here, we only focus on the discrete situation which is more related to our framework. Assuming we have two sets of points (features), the discrete distributions are formulated as:

$$U = \sum_{m=1}^{M} u_m \delta_{\boldsymbol{f}_m} \qquad \text{and} \qquad V = \sum_{n=1}^{N} v_n \delta_{\boldsymbol{g}_n}, \tag{2}$$

where $\boldsymbol{u}$ and $\boldsymbol{v}$ are the discrete probability vectors that sum to 1, and $\delta_{\boldsymbol{f}}$ is a Dirac delta function placed at support point $\boldsymbol{f}$ in the embedding space. Then, the total distance is written as:

$$< \boldsymbol{T}, \boldsymbol{C} > = \sum_{m=1}^{M} \sum_{n=1}^{N} \boldsymbol{T}_{m,n} \boldsymbol{C}_{m,n}. \tag{3}$$

We call $\boldsymbol{C}$ the cost matrix in which each point denotes the cost between $\boldsymbol{f}_m$ and $\boldsymbol{g}_n$, such as $\boldsymbol{C}_{m,n} = 1 - \text{sim}(\boldsymbol{f}_m, \boldsymbol{g}_n)$. While the $\boldsymbol{T}$ is called the transport plan, which is learned to minimize the total distance. The optimization problem of optimal transport is formulated as:

$$d_{\text{OT}}(\boldsymbol{u}, \boldsymbol{v}|\boldsymbol{C}) = \underset{\boldsymbol{T}}{\text{minimize}} < \boldsymbol{T}, \boldsymbol{C} >$$
$$\text{subject to} \quad \boldsymbol{T}\mathbf{1}_N = \boldsymbol{u}, \ \boldsymbol{T}^\top \mathbf{1}_M = \boldsymbol{v}, \ \boldsymbol{T} \in \mathbb{R}_+^{M \times N}. \tag{4}$$

As directly optimizing the above objective is always time-consuming, we apply the Sinkhorn distance (Cuturi, 2013) to use an entropic constraint for fast optimization. The optimization problem with a Lagrange multiplier of the entropy constraint is:

$$d_{\text{OT},\lambda}(\boldsymbol{u}, \boldsymbol{v}|\boldsymbol{C}) = \underset{\boldsymbol{T}}{\text{minimize}} < \boldsymbol{T}, \boldsymbol{C} > -\lambda h(\boldsymbol{T})$$
$$\text{subject to} \quad \boldsymbol{T}\mathbf{1}_N = \boldsymbol{u}, \ \boldsymbol{T}^\top \mathbf{1}_M = \boldsymbol{v}, \ \boldsymbol{T} \in \mathbb{R}_+^{M \times N}, \tag{5}$$

where $h(\cdot)$ is entropy and $\lambda \geq 0$ is a hyper-parameter. Then we can have a fast optimization solution with a few iterations as:

$$\boldsymbol{T}^* = \text{diag}(\boldsymbol{u}^{(t)}) \exp(-\boldsymbol{C}/\lambda) \text{diag}(\boldsymbol{v}^{(t)}), \tag{6}$$

where $t$ denotes the iteration and in each iteration $\boldsymbol{u}^{(t)} = \boldsymbol{u}/\left((\exp(-\boldsymbol{C}/\lambda)\boldsymbol{v}^{(t-1)}\right)$ and $\boldsymbol{v}^{(t)} = \boldsymbol{v}/\left((\exp(-\boldsymbol{C}/\lambda)^\top \boldsymbol{u}^{(t)}\right)$, with the initiation $\boldsymbol{v}^{(0)} = \mathbf{1}$.

## 3.3 PROMPT LEARNING WITH OPTIMAL TRANSPORT

In this subsection, we introduce the details of our **PLOT** , which learns multiple prompts to describe different characteristics of the category by minimizing the OT distance.

Specifically, as shown in Figure 2, given an image $\boldsymbol{x}$, we first feed it to the visual encoder branch of CLIP. Apart from the global visual feature $\boldsymbol{f}$, we can also obtain a set of local features $\{\boldsymbol{f}_m|_{m=1}^{M}\}$. The visual encoder has a multi-head attention pooling layer in which the input is the combination

of the global feature and a set of local features (feature map) and the output is a tensor with the shape $\mathbb{R}^{(H \times W + 1) \times C}$, where $H$ and $W$ is the height and width of feature map and $C$ is the feature dimension. Therefore, we can obtain $M = H \times W$ local features and a global feature. At the same time, for class $k$, we can initialize N local prompts as $\{t_{k,n}|_{n=1}^N\}$ with learnable vectors $\{\omega_n|_{n=1}^N\}$, where each is the same as the prompt in CoOp. With both visual and textual encoders, we can obtain local visual features $\boldsymbol{F} = \{\boldsymbol{f}_m|_{m=1}^M\} \in \mathbb{R}^{M \times C}$ and prompt features $\boldsymbol{G}_k = \{\boldsymbol{g}_n|_{n=1}^N\} \in \mathbb{R}^{N \times C}$.

In the inner loop, we learn the transport plan $\boldsymbol{T}$ with these fixed support sets $\boldsymbol{F}, \boldsymbol{G}_k$, by minimizing the following OT distance to push $\boldsymbol{G}_k$ to $\boldsymbol{F}$:

$$d_{\mathrm{OT}}(k) = d_{\mathrm{OT}}(\boldsymbol{u}, \boldsymbol{v}|\boldsymbol{1} - \boldsymbol{F}^\top \boldsymbol{G}_k), \tag{7}$$

where $\boldsymbol{C} = \boldsymbol{1} - \boldsymbol{F}^\top \boldsymbol{G}_k$ denotes that we use the cosine distance between $\boldsymbol{F}$ and $\boldsymbol{G}_k$ as the cost matrix. Then we can obtain the solution of transport plan $\boldsymbol{T}^*$ as Eq. 6 and the final OT distance $d_{\mathrm{OT}}(k)$.

Given the OT distance between $\boldsymbol{G}_k$ and $\boldsymbol{F}$, we reformulate the prediction probability as:

$$p_{\mathrm{OT}}(y = k|\boldsymbol{x}) = \frac{\exp\left((1 - d_{\mathrm{OT}}(k))/\tau\right)}{\sum_{k'=1}^K \exp\left((1 - d_{\mathrm{OT}}(k'))/\tau\right)}. \tag{8}$$

In the outer loop, we fix the transport plan $\boldsymbol{T}^*$ and optimize $\{\boldsymbol{vec}_{l,n}|_{l=1,n=1}^{L,N}\}$ with cross entropy:

$$L_{\mathrm{CE}} = -\frac{1}{|\mathcal{X}|} \sum_{\boldsymbol{x} \in \mathcal{X}} \sum_{k=1}^K y_{\boldsymbol{x},k} p_{\mathrm{OT}}(y = k|\boldsymbol{x}), \tag{9}$$

where $\boldsymbol{y}_{\boldsymbol{x}}$ is a one-hot label vector. The detailed algorithm can be found in Appendix A1.

Though the optimization strategy of the optimal transport and prompts is two-stage, the whole training flow is end-to-end. It is because the transport plan is computed using a small number of matrix multiplications as a forward module. The gradients of these matrix multiplications are taped for back-propagation for end-to-end optimization, which makes the whole system fully differentiable (including the iterative algorithm) and easy to implement using an autograd library like PyTorch. In the experiments, we found that it is natural and relatively easy to this optimization strategy.

## 4 EXPERIMENTS

Extensive experiments are conducted to evaluate our method, including comparison with CoOp, ablation studies, parameter analysis extensibility analysis, computing cost analysis, and visualization.

### 4.1 DATASETS

We followed the experimental settings in the CoOp (Zhou et al., 2021b) for the few-shot learning evaluation. The experiments are conducted on the 11 visual recognition datasets, including Caltech101 (Fei-Fei et al., 2004), DTD (Cimpoi et al., 2014), EuroSAT (Helber et al., 2019), FGV-CAircraft (Maji et al., 2013), Flowers102 (Nilsback & Zisserman, 2008), Food101 (Bossard et al., 2014), ImageNet (Deng et al., 2009), OxfordPets (Parkhi et al., 2012), StanfordCars (Krause et al., 2013), SUN397 (Xiao et al., 2010), and UCF101 (Soomro et al., 2012). These datasets span visual classification of generic objects, scenes, actions, fine-grained categories, and so on, which constitutes a comprehensive evaluation of our method. All experiments adopted the few-shot evaluation protocol used in CLIP (Radford et al., 2021) and CoOp (Zhou et al., 2021b), where we respectively choose 1, 2, 4, 8, and 16 shots for model training and use the original test set for evaluation. Besides, we also evaluated the robustness of our method with domain shift. Following CoOp, we used the ImageNet as the source domain and evaluate our method with ImageNet-based robustness evaluation datasets including ImageNetV2 (Recht et al., 2019), ImageNet-Sketch (Wang et al., 2019), ImageNet-A (Hendrycks et al., 2019), and ImageNet-R (Hendrycks et al., 2020). A detailed introduction of each dataset can be found in the appendix.

### 4.2 IMPLEMENTATION DETAILS

We chose CoOp (Zhou et al., 2021b) as our main competitor to evaluate our method. Compared with CoOp which only learns a global prompt for one class, our **PLOT** method learns multiple local prompts and applies the OT distance for fine-grained alignment. Besides, we also reported

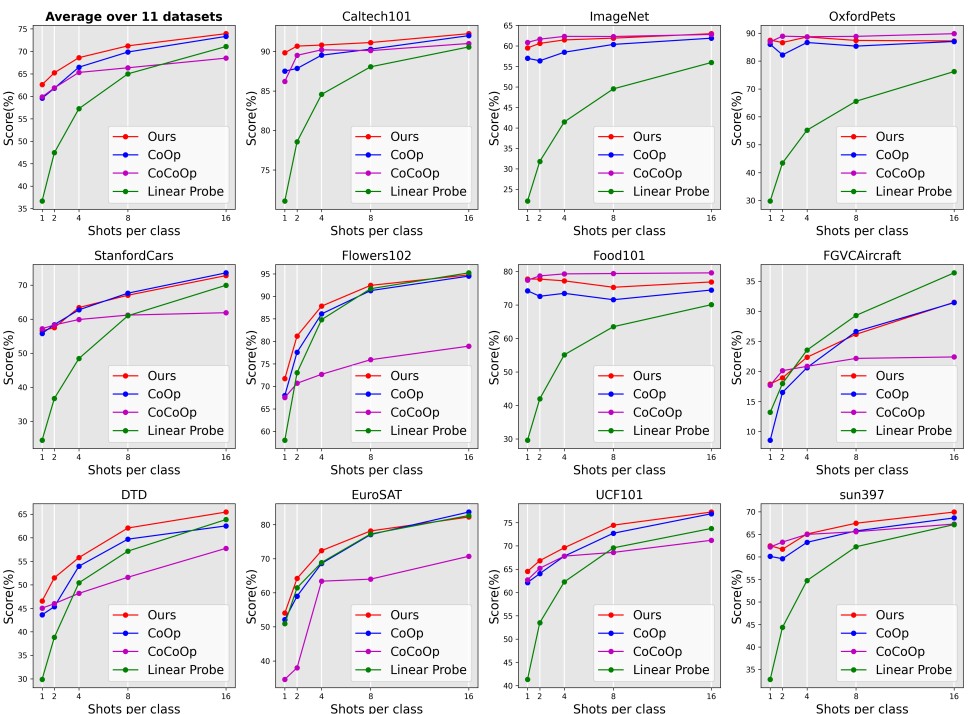

Figure 3: The few-shot learning results on 11 datasets. We compare our **PLOT** with CoOp, CoCoOp, and the Linear Probe method and observe the consistent and significant performance improvement on most datasets. (The average accuracy on all datasets is shown on the left top.)

the performance of training a linear classifier with the CLIP (Radford et al., 2021) features and the conditional version of CoOp, called CoCoOp (Zhou et al., 2022). They are also widely-used methods to adapt the pretrained knowledge for the downstream task. Please note that we evaluate CoCoOp in the same setting for a fair comparison (the base-to-new setting can be found in the appendix). The original CoOp method has different versions with different class token positions and parameter initialization strategies. For easy comparison, we directly chose one of them as our baseline with "end" token position, "random" initialization, 16 context tokens, and RN50 backbone. More implementation details can be found in Section A2.

## 4.3 COMPARISON WITH COOP

In this subsection, we compare our **PLOT** with the baseline CoOp on the few-shot recognition and domain generalization tasks.

**Few-Shot Learning** We summarized the experimental results in Figure 3 where the red line denotes our **PLOT** method, the blue one denotes CoOp, the purple line denotes CoCoOp (Zhou et al., 2022), and the green one is the CLIP linear probe. As the settings in the CoCoOp and CoOp are different, we re-run the CoCoOp method in the setting of CoOp. We observed that all prompt learning methods outperform the linear probe method by a large margin. Besides, **PLOT** can further outperform CoOp and CoCoOp on most of the datasets. Taking the average accuracy (at the left top) as the example, **PLOT** respectively gained $3.03\%, 3.45\%, 2.13\%, 1.38\%, 0.61\%$ performance boost over CoOp at $1, 2, 4, 8, 16$ shots. Among all datasets, **PLOT** achieves a larger improvement over CoOp on the FOOD101 and DTD datasets and achieves comparable performance on the StanfordCars datasets. It may be because the discriminative characters in StanfordCars coincide with each other, such that one global prompt and one global visual feature can work well. Note that we don't use the class-specific context, thus the performance on the fine-grained classification datasets is lower, e.g. the performance of both CoOp and **PLOT** without class-specific context is lower than the linear probing on FGVCAircraft. All these performance comparisons can serve as experimental evidence to demonstrate that multiple local prompts and the OT distance facilitate the prompt learning of vision-language models. The detailed accuracy can be found in the appendix.

Table 2: **Ablation studies on few-shot recognition**. `PLOT` : our defined model with $N = 4$. CoOp: the baseline method. G: respectively matching the global visual feature and multiple textual prompts V: applying a constraint to add the variance of prompts. E: using different initializations as the ensemble: M: using the visual feature map instead of the global visual feature. More details of different variants can be found in Section A2.4 in the appendix.

| Dataset | Settings | 1 shot | 2 shots | 4 shots | 8 shots | 16 shots |
|---|---|---|---|---|---|---|
| Caltech101 | **PLOT** | $89.83 \pm 0.33$ | $90.67 \pm 0.21$ | $90.80 \pm 0.20$ | $91.54 \pm 0.33$ | $92.24 \pm 0.38$ |
| | CoOp | $87.51 \pm 1.02$ | $87.84 \pm 1.10$ | $89.52 \pm 0.80$ | $90.28 \pm 0.42$ | $91.99 \pm 0.31$ |
| | G | $88.13 \pm 0.36$ | $86.98 \pm 1.25$ | $88.45 \pm 0.79$ | $90.16 \pm 0.22$ | $90.72 \pm 0.18$ |
| | G+V | $88.28 \pm 0.43$ | $87.72 \pm 1.25$ | $88.45 \pm 0.30$ | $89.82 \pm 0.20$ | $92.00 \pm 0.13$ |
| | G+E | $88.91 \pm 0.38$ | $90.01 \pm 0.22$ | $90.41 \pm 0.20$ | $90.60 \pm 0.10$ | $91.74 \pm 0.42$ |
| | M | $69.78 \pm 1.75$ | $71.57 \pm 1.59$ | $77.18 \pm 2.16$ | $81.77 \pm 0.47$ | $86.21 \pm 0.20$ |
| | M+V | $66.11 \pm 8.29$ | $71.45 \pm 3.98$ | $79.30 \pm 3.96$ | $86.96 \pm 0.78$ | $89.80 \pm 0.17$ |
| DTD | **PLOT** | $46.55 \pm 2.62$ | $51.24 \pm 1.95$ | $56.03 \pm 0.43$ | $61.70 \pm 0.35$ | $65.60 \pm 0.82$ |
| | CoOp | $43.62 \pm 1.96$ | $45.35 \pm 0.31$ | $53.94 \pm 1.37$ | $59.69 \pm 0.13$ | $62.51 \pm 0.25$ |
| | G | $45.12 \pm 1.69$ | $48.39 \pm 2.08$ | $54.75 \pm 0.48$ | $60.15 \pm 0.70$ | $63.59 \pm 0.76$ |
| | G+V | $45.90 \pm 2.00$ | $48.50 \pm 0.99$ | $53.96 \pm 0.48$ | $59.69 \pm 1.01$ | $63.51 \pm 0.66$ |
| | G+E | $46.39 \pm 1.00$ | $49.31 \pm 0.56$ | $52.99 \pm 0.60$ | $60.44 \pm 1.64$ | $63.97 \pm 0.48$ |
| | M | $13.18 \pm 4.57$ | $12.25 \pm 3.86$ | $13.00 \pm 4.73$ | $20.76 \pm 5.42$ | $26.99 \pm 1.98$ |
| | M+V | $12.61 \pm 5.93$ | $15.11 \pm 1.81$ | $20.35 \pm 1.33$ | $44.13 \pm 2.39$ | $56.85 \pm 0.54$ |
| FOOD101 | **PLOT** | $77.74 \pm 0.47$ | $77.70 \pm 0.02$ | $77.21 \pm 0.43$ | $75.31 \pm 0.30$ | $77.09 \pm 0.18$ |
| | CoOp | $74.25 \pm 1.52$ | $72.61 \pm 1.33$ | $73.49 \pm 2.03$ | $71.58 \pm 0.79$ | $74.48 \pm 0.15$ |
| | G | $74.63 \pm 0.11$ | $70.15 \pm 0.49$ | $70.41 \pm 0.46$ | $70.72 \pm 0.98$ | $73.68 \pm 0.46$ |
| | G+V | $74.83 \pm 0.31$ | $70.09 \pm 0.85$ | $70.86 \pm 0.22$ | $70.80 \pm 0.68$ | $73.93 \pm 0.35$ |
| | G+E | $75.77 \pm 0.62$ | $73.54 \pm 0.88$ | $75.82 \pm 0.44$ | $72.40 \pm 0.50$ | $75.52 \pm 0.33$ |
| | M | $52.02 \pm 4.86$ | $46.12 \pm 1.46$ | $46.86 \pm 1.39$ | $53.43 \pm 0.88$ | $61.28 \pm 0.23$ |
| | M+V | $46.52 \pm 1.15$ | $45.95 \pm 2.66$ | $53.57 \pm 0.83$ | $62.95 \pm 0.37$ | $67.63 \pm 1.11$ |

Table 3: Parameter analysis for the number of prompts

| Dataset | Settings | 1 shot | 2 shots | 4 shots | 8 shots | 16 shots |
|---|---|---|---|---|---|---|
| Caltech101 | N=1 | $88.47 \pm 1.15$ | $89.19 \pm 0.39$ | $89.70 \pm 0.38$ | $90.45 \pm 0.24$ | $91.56 \pm 0.14$ |
| | N=2 | $88.86 \pm 0.51$ | $89.60 \pm 0.10$ | $90.60 \pm 0.17$ | $91.25 \pm 0.65$ | $91.89 \pm 0.36$ |
| | N=4 | $\mathbf{89.83 \pm 0.33}$ | $\mathbf{90.67 \pm 0.21}$ | $90.80 \pm 0.20$ | $\mathbf{91.54 \pm 0.33}$ | $\mathbf{92.24 \pm 0.38}$ |
| | N=8 | $89.74 \pm 0.30$ | $90.18 \pm 0.46$ | $\mathbf{91.02 \pm 0.18}$ | $91.28 \pm 0.28$ | $92.04 \pm 0.29$ |
| DTD | N=1 | $43.91 \pm 0.65$ | $48.21 \pm 2.20$ | $53.69 \pm 1.10$ | $58.90 \pm 0.19$ | $62.85 \pm 0.74$ |
| | N=2 | $45.59 \pm 2.46$ | $48.06 \pm 1.92$ | $55.58 \pm 1.71$ | $61.56 \pm 0.17$ | $64.60 \pm 0.92$ |
| | N=4 | $46.55 \pm 2.62$ | $51.24 \pm 1.95$ | $\mathbf{56.03 \pm 0.43}$ | $61.70 \pm 0.35$ | $\mathbf{65.60 \pm 0.82}$ |
| | N=8 | $\mathbf{46.89 \pm 1.94}$ | $\mathbf{51.87 \pm 2.06}$ | $54.45 \pm 0.48$ | $\mathbf{62.20 \pm 0.56}$ | $65.25 \pm 0.38$ |
| FOOD101 | N=1 | $75.96 \pm 0.48$ | $76.12 \pm 0.59$ | $77.11 \pm 0.41$ | $76.56 \pm 0.69$ | $77.43 \pm 0.80$ |
| | N=2 | $77.12 \pm 0.49$ | $76.89 \pm 0.23$ | $76.16 \pm 0.52$ | $75.23 \pm 0.69$ | $76.81 \pm 0.50$ |
| | N=4 | $77.74 \pm 0.47$ | $77.70 \pm 0.02$ | $77.21 \pm 0.43$ | $75.31 \pm 0.30$ | $77.09 \pm 0.18$ |
| | N=8 | $\mathbf{78.05 \pm 0.15}$ | $\mathbf{78.19 \pm 0.07}$ | $\mathbf{78.12 \pm 0.17}$ | $\mathbf{76.63 \pm 0.22}$ | $\mathbf{77.48 \pm 0.12}$ |

**Domain generalization** The robustness also plays a critical role in model applications since the real-world environment may have large domain shifts with the training data. Therefore, we conducted a robustness evaluation to investigate the transferability of

Table 1: Comparisons on robustness to domain shift.

| Method | Source | Target | | |
|---|---|---|---|---|
| | ImageNet | -V2 | -Sketch | -A | -R |
| CLIP + CoOp | 61.91 | 54.26 | 32.47 | 21.78 | 54.21 |
| CLIP + `PLOT` ($N\!=\!4$) | **63.01** | **55.11** | **33.00** | **21.86** | **55.61** |

models learned by `PLOT` . Table 1 summarizes the results of our `PLOT` method and CoOp on four ImageNet-based robustness evaluation datasets. For both methods, we trained the models on ImageNet with 16 shots per class. For `PLOT` , we set the number of prompts as $N = 4$. We can observe that `PLOT` outperforms CoOp consistently on both source and target domains. These experimental results demonstrate that the performance improvement of our learning multiple prompts doesn't rely on single-domain overfitting.

## 4.4 ABLATION STUDIES AND MORE ANALYSIS

In this subsection, we conducted the ablation studies to investigate the effectiveness of different components, in order to answer the following questions.

Table 4: The few-shot accuracies of Tip-adapter-F and our adapter-based **PLOT** on 11 datasets.

| Dataset | Methods | 1 shot | 2 shots | 4 shots | 8 shots | 16 shots |
|---------|---------|--------|---------|---------|---------|----------|
| Caltech101 | Tip-Adapter-F + **PLOT** | 89.33 | 90.87 | 90.87 | 92.29 | 93.18 |
| | Tip-Adapter-F | 89.33 | 89.74 | 90.56 | 91.44 | 92.86 |
| DTD | Tip-Adapter-F + **PLOT** | 51.12 | 52.42 | 59.81 | 63.71 | 67.79 |
| | Tip-Adapter-F | 49.65 | 53.72 | 57.39 | 62.71 | 66.55 |
| EuroSAT | Tip-Adapter-F +**PLOT** | 64.37 | 76.53 | 79.51 | 79.17 | 85.75 |
| | Tip-Adapter-F | 59.53 | 66.15 | 74.12 | 77.93 | 84.54 |
| FGVCAircraft | Tip-Adapter-F +**PLOT** | 19.89 | 22.20 | 26.22 | 30.69 | 36.21 |
| | Tip-Adapter-F | 20.22 | 23.19 | 25.80 | 30.21 | 35.55 |
| Flowers102 | Tip-Adapter-F + **PLOT** | 77.59 | 84.98 | 89.32 | 93.75 | 96.10 |
| | Tip-Adapter-F | 79.98 | 82.30 | 88.83 | 91.51 | 94.80 |
| FOOD101 | Tip-Adapter-F + **PLOT** | 78.71 | 78.52 | 77.90 | 76.93 | 78.36 |
| | Tip-Adapter-F | 77.51 | 77.81 | 78.24 | 78.64 | 79.43 |
| ImageNet | Tip-Adapter-F + **PLOT** | 62.27 | 64.31 | 63.89 | 65.04 | 66.17 |
| | Tip-Adapter-F | 61.32 | 61.69 | 62.52 | 64.00 | 65.51 |
| OxfordPets | Tip-Adapter-F + **PLOT** | 87.16 | 87.68 | 88.63 | 89.78 | 87.54 |
| | Tip-Adapter-F | 87.00 | 87.03 | 87.54 | 88.09 | 89.70 |
| StanfordCars | Tip-Adapter-F + **PLOT** | 59.12 | 62.32 | 67.50 | 70.64 | 76.00 |
| | Tip-Adapter-F | 58.86 | 61.50 | 64.57 | 69.25 | 75.74 |
| SUN397 | Tip-Adapter-F + **PLOT** | 64.26 | 64.91 | 67.29 | 69.87 | 71.64 |
| | Tip-Adapter-F | 62.50 | 63.64 | 66.21 | 68.87 | 71.47 |
| UCF101 | Tip-Adapter-F + **PLOT** | 65.69 | 70.22 | 72.56 | 76.53 | 79.51 |
| | Tip-Adapter-F | 64.87 | 66.43 | 70.55 | 74.25 | 78.03 |
| Average | Tip-Adapter-F + **PLOT** | 65.45 | 68.63 | 71.23 | 73.49 | 76.20 |
| | Tip-Adapter-F | 64.62 | 66.65 | 69.67 | 72.45 | 75.83 |

**Q: Can we directly learn multiple prompts by matching the prompt ensemble with the global visual feature?  A: No**. As shown in Table 2, we report the performance of directly matching the prompt ensemble with the global visual feature (notated as "G") on three datasets including Caltech101, DTD, and FOOD101. The performance improvement of this method over CoOp is limited and far lower than **PLOT** . It may be because this "G" method is incentivized to learn the indistinguishable prompts, which contradicts our purpose to learn multiple comprehensive prompts.

**Q: Can ensemble methods that encourage the variety of prompts work well?  A: Not really**. As shown in Table 2, we further apply two methods to encourage the variety of prompts and then use the ensemble to match the global feature. In method "V", we add an objective function to add the distance between every two prompts as a regularization term. In method "E", we use predefined different initializations to replace the random initializations, such as "a photo of a", "this is a photo", "this is a", and "one picture of a". However, "G+V" did not achieve consistent improvement over the "G". Despite the clear improvement brought by "G+E", our **PLOT** showed consistent superiority over "G+E", which further demonstrates the effectiveness of the OT distance.

**Q: Does the improvement mainly come from using all feature maps? A: No**. In **PLOT** , we apply all feature maps of the visual encoder branch, where each feature is a local embedding at one spatial position. However, we demonstrate that the improvement of **PLOT** does not only rely on using all feature maps. On the contrary, directly using the feature map to replace the global feature causes a large performance drop. For example, on all three datasets, directly using the feature map ("M" or "M+V") has around 20% 1 shot accuracy drop over using the global visual feature. It is not surprising since the original CLIP model is trained by matching the global visual feature and language feature. Without using the OT method, the distance between the feature map and multiple textual prompts degenerates to the mean distance of each feature-prompt pair.

**Q: How many prompts are needed?  A: 4 prompts are enough** One important hyper-parameter in **PLOT** is the number of prompts. To analyze the effect of the number of prompts, we conducted the experiments on three datasets with $1, 2, 4, 8$ prompts. The results are summarized in the white part of Table 3. We can observe that the performance obviously increases when adding the number

Figure 4: Visualization. We provide the heatmaps of transport plan $T$ related to each prompt on 4 categories in ImageNet. Different transport plans focus on different attributes of the object.

of prompts from 1 to 4. For example, **PLOT** (N=4) respectively obtains $1.36\%$, $2.64\%$, and $1.68\%$ 1-shot accuracy improvement over **PLOT** (N=1) on three datasets. Besides, when we further increase the number of prompts, the improvement is not consistent. To balance the improvement and cost, we set $N = 4$ as the default configuration of our **PLOT** model. In the experiments, we tuned this hyper-parameter on the Caltech101 dataset and applied it to other datasets.

**Q: Can PLOT benefit Adapter-based methods?** **A: Yes**. Adapter-based methods (Gao et al., 2021; Zhang et al., 2021a) is another research direction of the efficient adaptation of pre-trained vision-language models. Different from prompt learning that fixes the model parameters and tunes the language prompt, adapter-based methods (Gao et al., 2021; Zhang et al., 2021a) allow for fine-tuning a part of the network or adding an extra model for training. Recently, adapter-based methods also achieved good performance on few-shot visual recognition. Therefore, we would like to explore whether our **PLOT** approach can benefit them, and how.

We apply the Tip-adapter-F (Zhang et al., 2021a) as our baseline method, which learns a $Linear(d, N_{cls} \times K_{shots})$ model to describe one image by the similarity with all training samples, where $d$ is the dimension of visual feature, $N_{cls}$ is the number of categories (e.g. 1000 in ImageNet), and $K_{shots}$ is the number of shots. Then, the final similarity consists of the original distance between the visual feature and prompt ensembling and the new distance calculated by the learned feature and one-hot vector of labels (whose dimension is $(N_{cls} \times K_{shots}, N_{cls})$). Please find details in Tip-adapter-F (Zhang et al., 2021a). To introduce **PLOT** to this framework, we first used the feature map to replace the global feature and then learned multiple linear models. As a result, with different local features and different linear models, we can obtain a $M \times N$ distance matrix and apply the Sinkhorn algorithm (Cuturi, 2013) to calculate the OT distance. Furthermore, we can apply the learned prompts as co-partner of the ensembling prompt to refine the final similarity. Table 4 summarizes the few-shot recognition results of the original Tip-Adapter-F method and our adapter-based **PLOT** methods on all 11 datasets.

**Q: Can PLOT benefit zero-shot learning?** **A: No**. The detailed analysis and discussions can be found in the appendix.

**Q: What is the extra computation time cost of PLOT over CoOp baseline? A: Around $10\%$ inference speed and $5\%$ training time**. Please see the detailed analysis in the appendix.

### 4.5 VISUALIZATION

In this subsection, we provide some visualization examples of the transport plans $T$ related to different prompts (N=4) in Figure 4. A detailed analysis of these visualization examples and further visualization results including the interpretation of the learned prompt, a T-SNE visualization of prompts, and the visualization of the false case can be found in Section A3

## 5 CONCLUSION

In this paper, we present a method, named **PLOT**, to learn multiple comprehensive prompts to describe diverse characteristics of one category. To avoid convergence to one point, we propose to apply the optimal transport to achieve the fine-grained alignment between both vision and language domains. We apply a two-stage optimization strategy where the inner loop fixes the prompts and learns the transport plan to calculate the cross-modality distance, and the outer loop uses this distance to optimize the prompt learner. We build our method on the base of CoOp and achieve significant improvement on the few-shot recognition task in various datasets, which demonstrates the advantage to learn multiple prompts instead of a single one.

## ACKNOWLEDGMENT

This project was partially supported by the National Institutes of Health (NIH) under Contract R01HL159805, by the NSF-Convergence Accelerator Track-D award #2134901, by a grant from Apple Inc., a grant from KDDI Research Inc, and generous gifts from Salesforce Inc., Microsoft Research, and Amazon Research.

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

*Appendix for*

# "PLOT: Prompt Learning with Optimal Transport for Vision-Language Models"

Appendix organization:

---

---

## A1    Method Details

The Optimal Transport (Monge, 1781) is initially introduced to find a transportation plan to move simultaneously several items at a minimal cost, such as moving a pile of sand to fill all the holes. Recently, it is widely used for the comparison of distributions. Mathematically, given two probability density function $U$ and $V$ over space $\mathcal{X}$ and $\mathcal{Y}$, the OT (Wasserstein) distance (Thorpe, 2019) can be defined as

$$D_{\text{OT}}(U, V) = \inf_{\Gamma} \int_{\mathcal{X} \times \mathcal{Y}} \boldsymbol{C}(\boldsymbol{x}, \boldsymbol{y}) d\gamma(\boldsymbol{x}, \boldsymbol{y}), \tag{10}$$

where $\boldsymbol{C}(\boldsymbol{x}, \boldsymbol{y})$ is the cost between two points in the space $\mathcal{X} \times \mathcal{Y}$, and $\Gamma$ denotes the set of transport plans between support points $\boldsymbol{x}$ and $\boldsymbol{y}$ (e.g. $\gamma(\boldsymbol{x}, \boldsymbol{y})$). We can regard two probability density functions $U$ and $V$ as piles and holes and $\boldsymbol{C}$ is the cost function of moving a unit of sand.

In our problem of multiple prompts learning, we formulate the sets of visual features and prompt features as two discrete distributions as

$$U = \sum_{m=1}^{M} u_m \delta_{\boldsymbol{f}_m} \qquad \text{and} \qquad V = \sum_{n=1}^{N} v_n \delta_{\boldsymbol{g}_n}, \tag{11}$$

where $\boldsymbol{u}$ and $\boldsymbol{v}$ are the discrete probability vectors that sum to 1, and $\delta_{\boldsymbol{f}}$ is a Dirac delta function placed at support point $\boldsymbol{f}$ in the embedding space. Given two support points $\boldsymbol{f}_m$ and $\boldsymbol{g}_n$, the cost function is written as $\boldsymbol{C}(\boldsymbol{f}_m, \boldsymbol{g}_n) = 1 - \text{sim}(\boldsymbol{f}_m, \boldsymbol{g}_n) = 1 - \frac{\boldsymbol{f}_m^\top \boldsymbol{g}_n}{||\boldsymbol{f}_m|| \cdot ||\boldsymbol{g}_n||}$. For simply, in this discrete situation, $\boldsymbol{C} \in \mathbb{R}^{M \times N}$ is a cost matrix in which each point denotes the cost between $\boldsymbol{f}_m$ and $\boldsymbol{g}_n$.

---

**Algorithm A1:** The training process of Prompt Learning with Optimal Transport

---

**Input:** Training few-shot image data: $\mathbf{X} = \{\boldsymbol{x}\}$, pretrained CLIP model $f$ and $g$, number of prompts $N$, entropy parameter $\lambda$, maximum number of iterations in inner and outer loops $T_{in}, T_{out}$.

**Output:** The parameters of prompts $\{\boldsymbol{\omega}_n|_{n=1}^N\}$

 1: Initialize $\{\boldsymbol{\omega}_n|_{n=1}^N\}$
 2: **for** $t_{out} = 1, 2, \ldots, T_{out}$ in the outer loop **do**
 3:      Obtain a visual feature set $\boldsymbol{F} \in \mathbb{R}^{M \times C}$ with the visual encoder $f(x)$;
 4:      Generate prompt feature set $\boldsymbol{G}_k \in \mathbb{R}^{N \times C}$ of each class with the textual encoder $\{g(t_k^n)\}|_{n=1}^N$;
 5:      Calculate the cost matrix $\boldsymbol{C}_k = \mathbf{1} - \boldsymbol{F}^\top \boldsymbol{G}_k \in \mathbb{R}^{M \times N}$ of each class
 6:      Calculate the OT distance with an inner loop: Initialize the $\boldsymbol{v}^{(0)} = \mathbf{1}$, $\delta = 0.01$ and $\Delta_v = \infty$
 7:      **for** $t_{in} = 1, 2, \ldots, T_{in}$ **do**
 8:         Update $\boldsymbol{u}^{(t_{in})} = \boldsymbol{u}/((\exp(-\boldsymbol{C}/\lambda)\boldsymbol{v}^{(t_{in}-1)})$
 9:         Update $\boldsymbol{v}^{(t_{in})} = \boldsymbol{v}/((\exp(-\boldsymbol{C}/\lambda)^\top \boldsymbol{u}^{(t_{in})})$
10:         Update $\Delta_v = \sum |\boldsymbol{v}^{(t_{in})} - \boldsymbol{v}^{(t_{in}-1)}|/N$
11:         **if** $\Delta_v < \delta$ **then**
12:            break
13:         **end if**
14:      **end for**
15:      Obtain optimal transport plan as $\boldsymbol{T}_k^* = \text{diag}(\boldsymbol{u}^{(t)}) \exp(-\boldsymbol{C}_k/\lambda)\text{diag}(\boldsymbol{v}^{(t)})$,
16:      Calculate the OT distance $d_{\text{OT}}(k) = <\boldsymbol{T}_k^*, \boldsymbol{C}_k>$
17:      Calculate the classification probability $p_{\text{OT}}(y = k|\boldsymbol{x})$ with the OT distance
18:      Update the parameters of prompts $\{\boldsymbol{\omega}_n|_{n=1}^N\}$ with cross-entropy loss $L_{\text{CE}}$
19: **end for**
20: **return** $\{\boldsymbol{\omega}_n|_{n=1}^N\}$

---

Then, the total distance of these two distributions is written as:

$$<\boldsymbol{T}, \boldsymbol{C}> = \sum_{m=1}^M \sum_{n=1}^N \boldsymbol{T}_{m,n} \boldsymbol{C}_{m,n}, \tag{12}$$

where the $\boldsymbol{T} \in \mathbb{R}^{M \times N}$ is a matrix of transport plan, which is learned to minimize the total distance. Each point $\boldsymbol{T}_{m,n}$ in $\boldsymbol{T}$ is a weight of local cost $\boldsymbol{C}_{m,n}$.

The optimization problem of optimal transport is formulated as:

$$d_{\text{OT}}(\boldsymbol{u}, \boldsymbol{v}|\boldsymbol{C}) = \underset{\boldsymbol{T}}{\text{minimize}} <\boldsymbol{T}, \boldsymbol{C}>$$
$$\text{subject to} \quad \boldsymbol{T}\mathbf{1}_N = \boldsymbol{u},\ \boldsymbol{T}^\top \mathbf{1}_M = \boldsymbol{v},\ \boldsymbol{T} \in \mathbb{R}_+^{M \times N}. \tag{13}$$

These constraints of $\boldsymbol{T}$ are used to match its marginal distributions and original discrete distributions in Eq. 11. In our framework, we treat visual features $\boldsymbol{f}_m$ and prompt features $\boldsymbol{g}_n$ equally and thus $\boldsymbol{u} = \mathbf{1}_{M \times 1}/M$ and $\boldsymbol{v} = \mathbf{1}_{N \times 1}/N$.

As directly optimizing the above objective is always time-consuming, we apply the Sinkhorn distance (Cuturi, 2013) to use an entropic constraint for fast optimization. The optimization problem with a Lagrange multiplier of the entropy constraint is:

$$d_{\text{OT},\lambda}(\boldsymbol{u}, \boldsymbol{v}|\boldsymbol{C}) = \underset{\boldsymbol{T}}{\text{minimize}} <\boldsymbol{T}, \boldsymbol{C}> -\lambda h(\boldsymbol{T})$$
$$\text{subject to} \quad \boldsymbol{T}\mathbf{1}_N = \boldsymbol{u},\ \boldsymbol{T}^\top \mathbf{1}_M = \boldsymbol{v},\ \boldsymbol{T} \in \mathbb{R}_+^{M \times N}, \tag{14}$$

where $h(\cdot)$ is entropy and $\lambda \geq 0$ is a hyper-parameter. Then we can have a fast optimization solution with a few iterations as:

$$\boldsymbol{T}^* = \text{diag}(\boldsymbol{u}^{(t)}) \exp(-\boldsymbol{C}/\lambda)\text{diag}(\boldsymbol{v}^{(t)}), \tag{15}$$

where $t$ denotes iteration and in each iteration $\boldsymbol{u}^{(t)} = \boldsymbol{u}/\left((\exp(-\boldsymbol{C}/\lambda)\boldsymbol{v}^{(t-1)})\right)$ and $\boldsymbol{v}^{(t)} = \boldsymbol{v}/\left((\exp(-\boldsymbol{C}/\lambda)^\top \boldsymbol{u}^{(t)})\right)$, with the initiation $\boldsymbol{v}^{(0)} = \mathbf{1}$. The detailed algorithms of the training and testing processes are shown in Algorithms A1 and A2

---

**Algorithm A2:** The inference process of Prompt Learning with Optimal Transport

---

**Input:** Testing image data: $\mathbf{X} = \{\boldsymbol{x}\}$, number of prompts $N$, number of classes $K$, learned prompts $\{\boldsymbol{t}_k^n|_{k=1,n=1}^{K,N}\}$, a frozen pretrained CLIP model including image encoder $f$ and text encoder $g$
**Output:** The classification of each image

1:  **for** $\boldsymbol{x}$ in $\mathbf{X}$ **do**
2:      Obtain a visual feature set $\boldsymbol{F} \in \mathbb{R}^{M \times C}$ with the visual encoder $f(x)$;
3:      Generate prompt feature set $\boldsymbol{G}_k \in \mathbb{R}^{N \times C}$ of each class with the textual encoder $\{g(t_k^n)\}|_{n=1}^N$;
4:      Calculate the cost matrix $\boldsymbol{C}_k = \boldsymbol{1} - \boldsymbol{F}^\top \boldsymbol{G}_k \in \mathbb{R}^{M \times N}$ of each class
5:      Calculate the OT distance with an inner loop: Initialize the $\boldsymbol{v}^{(0)} = \boldsymbol{1}$, $\delta = 0.01$ and $\Delta_v = \infty$
6:      **for** $t_{in} = 1, 2, \ldots, T_{in}$ **do**
7:          Update $\boldsymbol{u}^{(t_{in})} = \boldsymbol{u}/((\exp(-\boldsymbol{C}/\lambda)\boldsymbol{v}^{(t_{in}-1)})$
8:          Update $\boldsymbol{v}^{(t_{in})} = \boldsymbol{v}/((\exp(-\boldsymbol{C}/\lambda)^\top \boldsymbol{u}^{(t_{in})})$
9:          Update $\Delta_v = \sum |\boldsymbol{v}^{(t_{in})} - \boldsymbol{v}^{(t_{in}-1)}|/N$
10:         **if** $\Delta_v < \delta$ **then**
11:             break
12:         **end if**
13:     **end for**
14:     Obtain optimal transport plan as $\boldsymbol{T}_k^* = \text{diag}(\boldsymbol{u}^{(t)})\exp(-\boldsymbol{C}_k/\lambda)\text{diag}(\boldsymbol{v}^{(t)})$,
15:     Calculate the OT distance $d_{\text{OT}}(k) = <\boldsymbol{T}_k^*, \boldsymbol{C}_k>$
16:     Calculate the classification probability $p_{\text{OT}}(y = k|\boldsymbol{x})$ with the OT distance
17:     **return** $k^* = \max_k p_{\text{OT}}(y = k|\boldsymbol{x})$
18: **end for**

---

## A2  EXPERIMENTAL DETAILS

### A2.1  DATASET DETAILS

The datasets we used in the experiments follow CoOp (Zhou et al., 2021b), which include 11 datasets for few-shot visual recognition and 4 ImageNet-based datasets for generalization (robustness) evaluation. The details of each dataset are shown in Table A1, including the number of classes, the sizes of training and testing sets, and the original tasks.

### A2.2  IMPLEMENTATION DETAILS

The original CoOp method has different versions with different class token positions and parameter initialization strategies. As the performance gap among different versions is limited, we directly chose one of them as our baseline, where the token position is "end", the parameter initialization strategy is "random", and the length of learnable context tokens is set as 16. Following the widely used setting in (Zhou et al., 2021b; 2022; Gao et al., 2021; Zhang et al., 2021a), we also chose RN50 (He et al., 2016) as the backbone network of the visual branch. All the code of our method is based on CoOp, which adopted the SGD optimizer with 0.002 initial learning rate, CosineAnnealingLR schedule, and a warmup trick with 1e-5 learning rate. We also followed the epoch strategy to train more epochs for more shots. For small datasets such as FGVCAircraft, OxfordFlowers, and StanfordCars, the batch size is set as 32, while for the larger dataset such as Imagenet and SUN397, the batch size is set as 128.

We apply $N = 4$ prompts for each category and use $M = 7 \times 7$ due to the feature map size. We set the hyper-parameters in the Sinkhorn distances algorithm (Cuturi, 2013) as $\lambda = 0.1$ for all the datasets. We set the maximum iteration number of the inner loop as 100 and will early stop the iteration when the average absolute update value $\Lambda < 0.01$. We initialize all values in the vector $v$ and $\mu$ as $1/N$ and $1/M$ respectively. All models are conducted on the Pytorch (Paszke et al., 2019) 1.7.1 and trained on 4 NVIDIA A100 GPUs. We repeated the experiments three times with different seeds and reported the average.

Table A1: The detailed statistics of datasets used in experiments.

| Dataset | Classes | Training size | Testing size | Task |
|---------|---------|---------------|--------------|------|
| Caltech101 (Fei-Fei et al., 2004) | 100 | 4,128 | 2,465 | Object recognition |
| DTD (Cimpoi et al., 2014) | 47 | 2,820 | 1,692 | Texture recognition |
| EuroSAT (Helber et al., 2019) | 10 | 13,500 | 8,100 | Satellite image recognition |
| FGVCAircraft (Maji et al., 2013) | 100 | 3,334 | 3,333 | Fine-grained aircraft recognition |
| Flowers102 (Nilsback & Zisserman, 2008) | 102 | 4,093 | 2,463 | Fine-grained flowers recognition |
| Food101 (Bossard et al., 2014) | 101 | 50,500 | 30,300 | Fine-grained food recognition |
| ImageNet (Deng et al., 2009) | 1,000 | 1.28M | 50,000 | Object recognition |
| OxfordPets (Parkhi et al., 2012) | 37 | 2,944 | 3,669 | Fine-grained pets recognition |
| StanfordCars (Krause et al., 2013) | 196 | 6,509 | 8,041 | Fine-grained car recognition |
| SUN397 (Xiao et al., 2010) | 397 | 15,880 | 19,850 | Scene recognition |
| UCF101 (Soomro et al., 2012) | 101 | 7,639 | 3,783 | Action recognition |
| ImageNetV2 (Recht et al., 2019) | 1,000 | - | 10,000 | Robustness of collocation |
| ImageNet-Sketch (Wang et al., 2019) | 1000 | - | 50,889 | Robustness of sketch domain |
| ImageNet-A (Hendrycks et al., 2019) | 200 | - | 7,500 | Robustness of adversarial attack |
| ImageNet-R (Hendrycks et al., 2020) | 200 | - | 30,000 | Robustness of multi-domains |

Table A2: The few-shot visual recognition accuracy on 11 datasets.

| Dataset | Methods | 1 shot | 2 shots | 4 shots | 8 shots | 16 shots |
|---------|---------|--------|---------|---------|---------|----------|
| Caltech101 | **PLOT** | $89.83 \pm 0.33$ | $90.67 \pm 0.21$ | $90.80 \pm 0.20$ | $91.54 \pm 0.33$ | $92.24 \pm 0.38$ |
| | CoOp | $87.51 \pm 1.02$ | $87.84 \pm 1.10$ | $89.52 \pm 0.80$ | $90.28 \pm 0.42$ | $91.99 \pm 0.31$ |
| DTD | **PLOT** | $46.55 \pm 2.62$ | $51.24 \pm 1.95$ | $56.03 \pm 0.43$ | $61.70 \pm 0.35$ | $65.60 \pm 0.82$ |
| | CoOp | $43.62 \pm 1.96$ | $45.35 \pm 0.31$ | $53.94 \pm 1.37$ | $59.69 \pm 0.13$ | $62.51 \pm 0.25$ |
| EuroSAT | **PLOT** | $54.05 \pm 5.95$ | $64.21 \pm 1.90$ | $72.36 \pm 2.29$ | $78.15 \pm 2.65$ | $82.23 \pm 0.91$ |
| | CoOp | $52.12 \pm 5.46$ | $59.00 \pm 3.48$ | $68.61 \pm 3.54$ | $77.08 \pm 2.42$ | $83.69 \pm 0.47$ |
| FGVCAircraft | **PLOT** | $17.90 \pm 0.09$ | $18.94 \pm 0.44$ | $22.36 \pm 0.42$ | $26.17 \pm 0.29$ | $31.49 \pm 0.89$ |
| | CoOp | $8.59 \pm 5.79$ | $16.52 \pm 2.38$ | $20.63 \pm 2.46$ | $26.63 \pm 0.86$ | $31.43 \pm 0.96$ |
| Flowers102 | **PLOT** | $71.72 \pm 0.97$ | $81.19 \pm 0.79$ | $87.82 \pm 0.20$ | $92.43 \pm 0.25$ | $94.76 \pm 0.34$ |
| | CoOp | $67.98 \pm 1.98$ | $77.58 \pm 1.46$ | $86.10 \pm 1.05$ | $91.27 \pm 0.83$ | $94.49 \pm 0.40$ |
| FOOD101 | **PLOT** | $77.74 \pm 0.47$ | $77.70 \pm 0.02$ | $77.21 \pm 0.43$ | $75.31 \pm 0.30$ | $77.09 \pm 0.18$ |
| | CoOp | $74.25 \pm 1.52$ | $72.61 \pm 1.33$ | $73.49 \pm 2.03$ | $71.58 \pm 0.79$ | $74.48 \pm 0.15$ |
| ImageNet | **PLOT** | $59.54 \pm 0.16$ | $60.64 \pm 0.06$ | $61.49 \pm 0.23$ | $61.92 \pm 0.09$ | $63.01 \pm 0.13$ |
| | CoOp | $56.99 \pm 1.03$ | $56.40 \pm 0.87$ | $58.48 \pm 0.47$ | $60.39 \pm 0.57$ | $61.91 \pm 0.17$ |
| OxfordPets | **PLOT** | $87.49 \pm 0.57$ | $86.64 \pm 0.63$ | $88.63 \pm 0.26$ | $87.39 \pm 0.74$ | $87.21 \pm 0.40$ |
| | CoOp | $85.99 \pm 0.28$ | $82.22 \pm 2.15$ | $86.65 \pm 0.97$ | $85.36 \pm 1.00$ | $87.02 \pm 0.89$ |
| StanfordCars | **PLOT** | $56.60 \pm 0.36$ | $57.52 \pm 0.71$ | $63.41 \pm 0.29$ | $67.03 \pm 0.50$ | $72.80 \pm 0.75$ |
| | CoOp | $55.81 \pm 1.67$ | $58.41 \pm 0.43$ | $62.74 \pm 0.16$ | $67.64 \pm 0.06$ | $73.60 \pm 0.19$ |
| SUN397 | **PLOT** | $62.47 \pm 0.43$ | $61.71 \pm 0.65$ | $65.09 \pm 0.43$ | $67.48 \pm 0.04$ | $69.96 \pm 0.24$ |
| | CoOp | $60.12 \pm 0.82$ | $59.60 \pm 0.76$ | $63.24 \pm 0.63$ | $65.77 \pm 0.02$ | $68.36 \pm 0.66$ |
| UCF101 | **PLOT** | $64.53 \pm 0.70$ | $66.83 \pm 0.43$ | $69.60 \pm 0.67$ | $74.45 \pm 0.50$ | $77.26 \pm 0.64$ |
| | CoOp | $62.13 \pm 1.14$ | $64.05 \pm 0.99$ | $67.79 \pm 0.71$ | $72.71 \pm 0.50$ | $76.90 \pm 0.50$ |
| Average | **PLOT** | $62.59 \pm 1.13$ | $65.23 \pm 0.72$ | $68.60 \pm 0.52$ | $71.23 \pm 0.51$ | $73.94 \pm 0.54$ |
| | CoOp | $59.56 \pm 2.06$ | $61.78 \pm 1.39$ | $66.47 \pm 1.29$ | $69.85 \pm 0.69$ | $73.33 \pm 0.42$ |

## A2.3 FEW-SHOT RECOGNITION ACCURACY

In Section 4.3.1, we provide a line chart to show and compare the performance of **PLOT** and CoOp. Here, we provide detailed performance results on all 11 few-shot recognition datasets in Table A2, where we use gray for our method and white for CoOp. To highlight, we respectively use dark cyan and light cyan to represent the performance of **PLOT** and CoOp on the average of all 11 datasets. We repeat all experiments 3 times and report the mean and standard deviation in the table.

## A2.4 ABLATION STUDIES DETAILS

In this section, we provide more details about the different variants in Table 2. We compare **PLOT** with the other 6 baseline methods briefly described below:

- CoOp: CoOp is the baseline method that only learns a single prompt and matches this single prompt and the global visual feature. We apply the officially released code to reproduce this method.
- "G": In this paper, we propose to explore whether we can learn multiple prompts for more comprehensive textual representation and fine-grained visual-textual alignment. "G" denotes that we build multiple prompts (similar to our **PLOT** ) and learn them by matching them with the single global visual feature.
- "G+V": Matching all local prompts to a single visual feature will reduce the diversity of the learned prompts. To improve the variety of learned prompts, "G+V" further adds an objective function to increase the distances between every two prompts.
- "G+E": "G+E" is also a method to increase the variety of prompts by separated initializations. It applies predefined different initializations to replace the random initialization, such as "a photo of a", "this is a photo", "this is a", and "one picture of a".
- "M": One key difference between **PLOT** and CoOp is to utilize the feature map for more fine-grained information. To evaluate whether our improvement mainly comes from using a feature map, we design a method "M", which removes the OT distance of **PLOT** and matches local visual features and multiple textual prompts by the average distance of each visual-textual pair.
- "M+V": Similar to "G+V", we add an objective function to increase the distances between every two prompts to the method "M" to increase the variety of prompts.

## A2.5 BASE-TO-NEW RESULTS

To investigate the generalization of our method for other baseline prompt-learning-based methods, we apply our **PLOT** to CoCoOp (Zhou et al., 2022), by learning multiple textual prompts (e.g. N=4) instead of the single prompt in CoCoOp. We name it **CoPLOT**. Specially, we learn multiple prompts and use the same meta-network for all local prompts. Then we apply the Optimal Transport to calculate the distance between multiple local prompts and local visual features. We evaluate both CoCoOp and CoPLOT in the setting of "base-to-new" and implement them using the same RN50 backbone. The results on the 11 datasets with 16 shots are provided in Table A3. We observe that PLOT achieves improvement on most datasets and on average, which demonstrates that it can be applied to different prompt-learning-based methods. For example, on average, **PLOT** achieves almost 3% improvement on the "new" side without reducing "base" performance. It suggests that these two methods are complementary: CoCoOp proposes a conditional formulation that uses each image feature as the context condition to refine the single prompt, while PLOT aims to learn multiple prompts.

## A2.6 ZERO-SHOT SETTING ANALYSIS

**PLOT** can not benefit in the setting of zero-shot learning. Below we provide some experimental details and corresponding analysis. CLIP shows that manually designing the prompts can still achieve good performance. We obtain 7 prompts by prompt engineering on the ImageNet dataset and can further ensemble them to obtain **60.38%** top 1 accuracy. In this section, we replace the cosine distance between the global visual feature and prompt ensemble with the OT distance between the feature map and all 7 prompts. However, without any learning, the OT distance only obtains **58.78%** accuracy. It is a limitation of the **PLOT** to still need few-shot data for optimization, which cannot be directly applied in the zero-shot setting. We argue there are two reasons why the OT distance does not work without learning: 1) prompt engineering selects prompts based on the global feature and cosine distance, instead of OT distance with feature map; 2) all these selected prompts are close to the global feature and lack the complementarity.

Table A3: **Comparison of CoCoOp (Zhou et al., 2022) and CoPLOT(ours) in the base-to-new generalization setting**. All methods are implemented with RN50 backbone and evaluated with 16 shots. We report the performance of the base classes, new classes, and the mean of them. We show that PLOT can be applied to CoCoOp and achieve improvement.

(a) **Average** .

|  | Base | New | H |
|---|---|---|---|
| CoCoOp | 75.7 | 64.6 | 70.2 |
| CoPLOT | 75.9 | 67.6 | 71.8 |

(b) ImageNet.

|  | Base | New | H |
|---|---|---|---|
| CoCoOp | 68.3 | 63.1 | 65.7 |
| CoPLOT | 68.2 | 63.1 | 65.7 |

(c) Caltech101.

|  | Base | New | H |
|---|---|---|---|
| CoCoOp | 95.0 | 90.0 | 92.5 |
| CoPLOT | 95.4 | 90.9 | 93.2 |

(d) OxfordPets.

|  | Base | New | H |
|---|---|---|---|
| CoCoOp | 92.3 | 94.6 | 93.5 |
| CoPLOT | 92.1 | 95.9 | 94 |

(e) StanfordCars.

|  | Base | New | H |
|---|---|---|---|
| CoCoOp | 61.8 | 65.3 | 63.6 |
| CoPLOT | 63.2 | 66.5 | 64.9 |

(f) Flowers102.

|  | Base | New | H |
|---|---|---|---|
| CoCoOp | 91.2 | 67.5 | 79.4 |
| CoPLOT | 89.6 | 69.2 | 79.4 |

(g) Food101.

|  | Base | New | H |
|---|---|---|---|
| CoCoOp | 85.0 | 86 | 85.5 |
| CoPLOT | 85.0 | 85.2 | 85.1 |

(h) FGVCAircraft.

|  | Base | New | H |
|---|---|---|---|
| CoCoOp | 25.5 | 25.7 | 25.6 |
| CoPLOT | 25.6 | 26.6 | 26.1 |

(i) SUN397.

|  | Base | New | H |
|---|---|---|---|
| CoCoOp | 75.1 | 73.6 | 74.4 |
| CoPLOT | 75.2 | 73.2 | 74.2 |

(j) DTD.

|  | Base | New | H |
|---|---|---|---|
| CoCoOp | 73.1 | 50.0 | 61.6 |
| CoPLOT | 72.6 | 51.4 | 62.0 |

(k) EuroSAT.

|  | Base | New | H |
|---|---|---|---|
| CoCoOp | 88.9 | 33.5 | 61.2 |
| CoPLOT | 91.0 | 55.3 | 73.2 |

(l) UCF101.

|  | Base | New | H |
|---|---|---|---|
| CoCoOp | 76.5 | 61.6 | 69.1 |
| CoPLOT | 77.4 | 66.2 | 71.8 |

Table A4: The training and inference time comparison.

| Settings | CoOp | **PLOT** (N=1) | **PLOT** (N=2) | **PLOT** (N=4) | **PLOT** (N=8) |
|---|---|---|---|---|---|
| Training Time (s) | 1.127 | 1.135 | 1.148 | 1.182 | 1.267 |
| Inference Time (images/s) | 719.1 | 714.4 | 690.7 | 653.0 | 519.8 |

### A2.7 COMPUTATION COST EVALUATION

As shown in Table A4, we provide the comparison of the training time and inference seed of the baseline method CoOp (Zhou et al., 2021b) and our **PLOT** with the different number of prompts. We report the one-epoch time training on the 1-shot setting of the Food101 (Bossard et al., 2014) dataset and the number of images processed by the model in 1 second. Taking $N = 4$ as an example, **PLOT** only reduces the $9.2\%$ inference speed and requires an extra $4.9\%$ training time, which is acceptable given the performance improvement.

## A3 VISUALIZATION

### A3.1 MORE ANALYSIS ON VISUALIZATION

In this section, we provide some visualization examples of the transport plans $T$ related to different prompts (N=4). We translate each transport plan into colorful heatmaps and resize them to their original size and combine them with the raw image. As shown in Figure 4, we provide the heatmaps of 4 categories in ImageNet. We observe that different transport plans highlight different regions of the image, which demonstrates that the learned multiple prompts are complementary. For the class "Brambling", the prompts respectively focus on the head, tail, wing, and environment. For "Dog Sled", the prompts are related to dogs, the sled, some ties, and the snow environment.

Table A5: The nearest words for 16 context vectors of all $N = 4$ prompts learned by **PLOT** . N/A means non-Latin characters.

| Number | Prompt 1 | Prompt 2 | Prompt 3 | Prompt 4 |
|---|---|---|---|---|
| 1 | ag | pa | trying | gaz |
| 2 | flint | as | field | **white** |
| 3 | leaving | wit | N/A | t |
| 4 | sot | l | icons | ario |
| 5 | tint | N/A | eclub | safe |
| 6 | tar | yl | indiffe | class |
| 7 | attn | N/A | ts | represented |
| 8 | 2 | job | cold | attend |
| 9 | rollingstones | built | yeah | vie |
| 10 | N/A | brought | band | recognized |
| 11 | N/A | or | love | old |
| 12 | bel | j | late | stel |
| 13 | **head** | ag | industry | awhile |
| 14 | artifact | bad | N/A | ded |
| 15 | an | chie | across | these |
| 16 | 5 | in | actual | visiting |

## A3.2 VISUALIZATION OF FAILURE CASES

To better understand the method and further discover the reason for the failure cases. we visualize the attention maps of some failure cases. As shown in Figure A1, we showed two failure examples with class "2000 AM General Hummer" in the StanfordCars dataset. During the training, we set the number of prompts as 4, but in these visualization results, we found that some of the learned prompts remarkably coincide with each other. These prompts can be roughly divided into two classes: Foreground and Background. For example, in both images, prompts 2 (right top) and 3 (left down) focus on the foreground car, while the others focus on the background. It demonstrates that not all classes have multiple complementary attributes, which motivates us to go further to learn the dynamic local prompts numbers to reduce the computational load in the future.

## A3.3 INTERPRETATION OF TEXT PROMPTS

The learned prompts are difficult to be understood by humans since the parameters are optimized in the continuous space (Zhou et al., 2021b). CoOp proposes to use the word which is nearest to learned prompts in the embedding space to visualize the prompts. Following this manner, we show the nearest words of our learned prompts in Table A5. Similar to CoOp, most words can not be directly understood by human logic. However, we still find the relations between the learned prompts and the corresponding optimal transport plan. As shown in Figure 4 in the main paper, we can observe that the optimal transport plan for Prompt 1 always focuses on the "head", such as the head of "brambling", the head of "rooster", and even the head of "aircraft carrier". It is because the word "head" is in Prompt 1. Similarly, we can find that Prompt 4 prefers the white part of images, such as the white environment in the image of "brambling" and the snow in the image of "dog sled". It demonstrates that the learned multiple prompts focus on different characteristics of categories.

## A3.4 T-SNE OF PROMPTS

To better understand the learned prompts, we provide a visualization with T-SNE (Van der Maaten & Hinton, 2008) for the learned textual prompts. Specifically, we randomly select 10 classes from ImageNet and generate the textual embedding with our learned prompts. Then, we obtain $4 \times 10$ embeddings with dimension $d = 1024$. Then we apply the T-SNE to reduce the dimension and visualize the embeddings. As shown in Figure A2, the textual embeddings of the same class with different prompts are clustered well. Besides, despite being well clustered, we found that the textual embeddings also have intra-diversities.

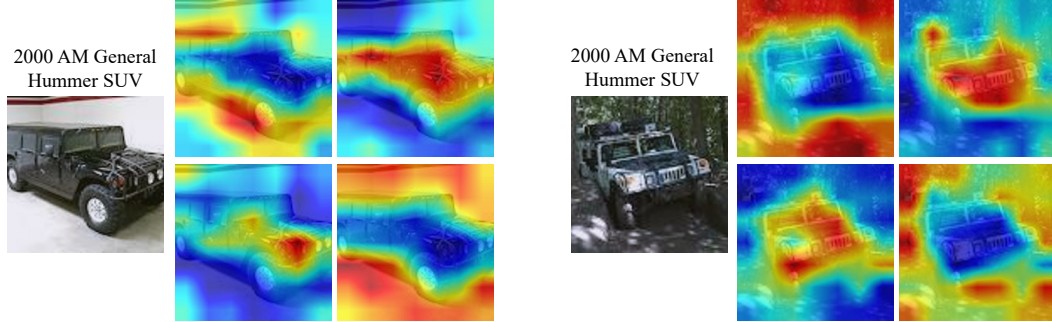

Figure A1: Failure Visualization. We provide the heatmaps of transport plan T related to each prompt on 2 failure examples in the StanfordCars dataset.

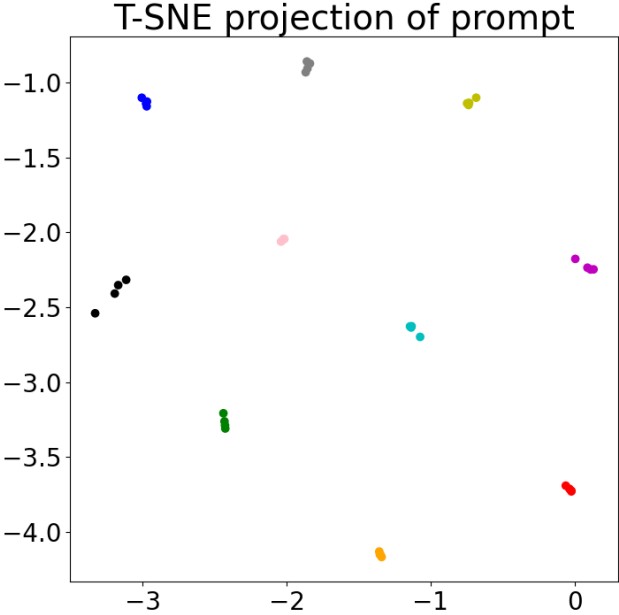

Figure A2: T-SNE Visualization of 10 classes with different prompts. We apply the T-SNE for the embeddings of 10 randomly selected classes in ImageNet with different prompts. Different color denotes different classes. We observe that the textual embeddings cluster well.

