# OpenReview forum: "PLOT: Prompt Learning with Optimal Transport for Vision-Language Models"
_ICLR.cc/2023/Conference — ICLR 2023 notable top 25%_

### Official Review · Reviewer_RWPs · 2022-10-13

**Confidence:** 4
**Correctness:** 4
**Technical Novelty And Significance:** 3
**Empirical Novelty And Significance:** 3
**Recommendation:** 8

**Clarity, Quality, Novelty And Reproducibility:**

The method is novel and the code is provided, hence reproducibility and novelty are properly covered in the manuscript.

The paper could benefit from better writing and clarity in the presentation; in my humble opinion, writing the paper top to bottom (i.e. presenting first optimal transport and then the problem) is more confusing than writing it bottom to top (i.e. the problem of prompt learning and the intuition behind the optimal transport in ''assigning" prompts to different visual features from the same image, there is barely discussion around it).

**Strength And Weaknesses:**

Strengths:

The paper is technically solid and contributes enough novelty to the domain of prompt learning which is of growing interest to the research community devoted to exploring pre-trained vision/language models. The use of a diverse set of prompts to generate different text classifiers that can all contribute to a single class prediction is of interest and the methodology is well developed.

The results seem to consistently outperform CoOp, showing the need of having a large set of prompts that require proper use to be able to generalize better to new classes.

Weaknesses:

While the core methodology is new, the idea of learning a distribution of prompts is not new, and indeed the authors do not include a discussion on why their proposed approach would be a better choice than CoCoOp which produces prompts for each image. Additionally, the authors need to consider, discuss, and compare, against

[Lu et al.] Prompt Distribution Learning. CVPR 2022

which is a method that learns a collection of prompts that produce a distribution of per-class weights, increasing their expressivity. Some analysis in terms of methodology and computation against [Lu et al.] should be included in the paper in my opinion.

**Summary Of The Paper:**

The paper proposes a method for prompt learning in vision and language models that aims at producing a variety of templates that can be used by the text encoder to provide better classification weights in a few-shot scenario.

In particular, the paper observes that learning a single prompt is not sufficient to represent a class, and proposes a variety of prompt candidates that can be used to cover a wider variety of concepts within the same image. The authors note that using a distribution of prompts that are learned independently using a standard contrastive learning will result in some of the prompts to collapse to the same centroid, and propose an optimal transport scheme to avoid such behaviour.

The methodology goes as follows: 1) a set of different visual features are computed for a given image (using different backbone exits); then 2) the scores for the pairs visual/text features are computed using the learnable prompts; then 3) a transport matrix is computed to minimize the total cost of assigning a feature vector a weighted combination of the text features (i.e. the transport matrix is the soft-extension of the permutation matrix that would assign vision/text features through the typical CLIP matching). Finally, for a fixed transport matrix 4) the prompts are learned using the standard cross-entropy where the class assignment is given by the computed combination of the weights produced by the prompts.

The use of a transport matrix allows to assign to each feature vector a weighted combination of the prompts, thus making the latter have a broader "semantic" capacity. The use of different feature vectors for each image rather than a global CLIP-based visual feature vector allows the prompt to describe different concepts that might co-occur in the same image, making the prompts having more ``local" capacity.

The paper is validated in the same setting as CoOp for few-shot learning, using 11 visual recognition datasets, delivering consistently better results than the competing method.


**Summary Of The Review:**

In summary, the paper proposes an interesting approach to prompt learning that is technically sound and well motivated. However, the clarity and the lack of an exhaustive comparison against state of the art works ([Lu et al.] and CoCoOp) both experimentally and conceptually is needed. Should the authors be able to bring this up would further increase the quality of the paper.

---

> ### Author Response · Authors · 2022-11-18
> **Response to Reviewer RWPs**
>
> Dear Reviewer RWPs, we greatly appreciate your valuable questions and helpful suggestions! We have added more discussions in the revised paper, according to your suggestions. Please see the revised paper and appendix for details.
>
> > **Q1:** While the core methodology is new, the idea of learning a distribution of prompts is not new. Additionally, the authors need to consider, discuss, and compare, against PDL [Lu et al.], which is a method that learns a collection of prompts that produce a distribution of per-class weights, increasing their expressivity. Some analysis in terms of methodology and computation against PDL [Lu et al.] should be included in the paper in my opinion.
>
> **A1:** Thank you very much for recommending a closely related, excellent recent work PDL [Lu et al.]. We have included a discussion with PDL in the related work of our revised paper. As you noticed, both learning distribution (Lu et al., 2022) and learning multiple prompts (our work) are motivated by more diverse prompts, the core methodologies are different: learning distribution of prompts predefines a parametric distribution and fits the parameters of this distribution during training; while PLOT learns multiple prompts without parametric distribution (it can be regarded as non-parametric discrete single-point distribution) and utilizes multiple local features in the visual feature map to learn an Optimal Transport distance for better fine-grained visual-textual alignment.
>
> The empirical comparisons show that PDL achieves better performance than ours.
> It is worth noting that their contribution and our design are complementary.  We simply leveraged both of them to learn multiple distributions, which further improves the performance, e.g. 91.2 -> 93.3 in the Caltech101 dataset (Please kindly refer to Section A7 in the appendix for details). This also suggests that the two contributions are somehow orthogonal.
>
>
> > **Q2:** indeed the authors do not include a discussion on why their proposed approach would be a better choice than CoCoOp which produces prompts for each image.
>
> **A2:** Thanks for this question. We have included a discussion with CoCoOp in Section A6 of the appendix.
> CoCoOp proposes a conditional formulation that uses each image feature as the context condition to refine the single prompt, while PLOT aims to learn multiple prompts. For example, given an image and its label, CoCoOp learns the single prompt of the textual label and uses the image feature to refine it, while PLOT learns multiple prompts and matches these local prompts with local visual features of the image. These two methods are complementary: PLOT aims to learn comprehensive prompts while CoCoOp attempts to learn conditional prompts. We have provided an experiment in which we can combine PLOT and CoCoOp to learn better prompts. Please kindly refer to Section A6 of the appendix for the details.
>
> Hope these responses can address your concerns.
>
> With best regards,
>
> Authors of submission 898

---

### Official Review · Reviewer_1oAQ · 2022-10-22

**Confidence:** 4
**Correctness:** 3
**Technical Novelty And Significance:** 3
**Empirical Novelty And Significance:** 3
**Recommendation:** 8

**Clarity, Quality, Novelty And Reproducibility:**

Overall, I think the presentation of this paper is clear and easy to understand. The proposed method is novel and technically sound.

**Strength And Weaknesses:**

__Strength__

1. This paper is overall well-structured and easy to follow.

2. The idea of learning multiple comprehensive prompts to describe the image categories is reasonable to me, and using optimal transport for optimization prompt learning is novel.

3. The ablation studies are comprehensive which clearly demonstrate the design choices of the proposed method.

__Weakness__

1. In Figure 3, the improvement achieved by the proposed method over CoCoOp is marginal. I wonder if the conditional formulation presented in CoCoOp can be easily extended to PLOT? If so, at least more detailed discussions should be provided by the authors.

2. Figure 4 shows the visualization results of attended regions of images from different prompts. It is also interesting to see how learned inter-class and intra-class prompts are different from each other by using t-sne or PCA.

3. It is unclear to see the differences among model variants in Table 2 just from the caption. It will be good if an individual section on introducing the detailed design/training setup of each model in the paper (or in the appendix).


**Summary Of The Paper:**

This paper presented a new prompt learning approach for vision-language models, namely PLOT. The main idea of this paper is to learn multiple prompts per class and use optimal transport to match them with visual features of images. The whole optimization process contains inner and outer loops to optimize the matching and prompts, respectively. The experiment results show that the proposed approach can achieve state-of-the-art performance in the few-shot image recognition task with a slight increasement in training time.

**Summary Of The Review:**

This paper addressed prompt learning for vision-language models which is an interesting and important problem in few-shot image classification tasks. The proposed PLOT is reasonable and technically sound to me. However, I still have some concerns on the experiment part of the paper. Therefore, it leads me to borderline acceptance as my initial rating. I will be happy to increase my rating if the concerns could be addressed in the discussion period.

---

> ### Author Response · Authors · 2022-11-18
> **Response to Reviewer 1oAQ (Part I)**
>
> Dear Reviewer 1oAQ, we sincerely appreciate your informative feedback and helpful suggestions that helped improve our paper. We have added the new experiments following your suggestions and modified the paper and appendix accordingly. Please see our point-to-point response below.
>
> > **Q1:** In Figure 3, the improvement achieved by the proposed method over CoCoOp is marginal. I wonder if the conditional formulation presented in CoCoOp can be easily extended to PLOT? If so, at least more detailed discussions should be provided by the authors.
>
> **A1:** We appreciate this helpful suggestion.  The marginal improvement of PLOT over CoCoOp in Figure 3 is because that PLOT uses the CoOp as the baseline which does not use the meta-network to learn the context condition (the conditional formulation presented in CoCoOp). In light of your suggestion to better compare our method with CoCoOp and evaluate whether CoCoOp can be easily extended to "CoPLOT", we apply the PLOT to CoCoOp to learn multiple prompts instead. Specifically, we build multiple learnable prompts and apply the same condition to each local prompt (i.e. it means that we don't change the meta-network model and share the outputs of the meta-network to all local prompts). Then, similar to PLOT, we calculate the distance between multiple local prompts and local visual features by the Optimal Transport distance to match the visual and textual features.  We evaluate both CoCoOp and CoPLOT in the setting of "base-to-new" and implement them using the same RN50 backbone. The results and analysis on the 11 datasets with 16 shots are provided in Table A4 and Section A6 of the appendix and shown in the table below. On average, it achieves 3% performance improvement on the "new" side without the reduction of the "base" performance. We observe that PLOT achieves improvement on most datasets, which demonstrates that it can be applied to different prompt-learning-based methods.
>
> | Method |Average |  Caltech  | DTD |  EuroSAT | Food101 | Flowers | FGVC | ImageNet | OxfordPets | StanfordCars  |  SUN397  |  UCF101  |
> |-------|:-------:|:-------:|:-------:|:-------:|:-------:|:-------:|:-------:|:-------:|:-------:|:-------:|:-------:|:-------:|
> | CoCoOp  | 75.7/64.6 |95.0/90.0 | 73.1/50.0 | 88.9/33.5 |  85.0/86.0 | 91.2/67.5 | 25.5/25.7 | 68.3/63.1 | 92.3/94.6 | 61.8/65.3  | 75.1/73.6 | 76.5/61.6 |
> | CoPLOT| 75.9/67.6 | 95.4/90.9 | 72.6/51.4 | 91.0/55.3 |  85.0/85.2 | 89.6/69.2 | 25.6/26.6 | 68.2/63.1 | 92.1/95.9 | 63.2/66.5  |75.2/73.2 | 77.4/66.2 |
>
>
> > **Q2:** Figure 4 shows the visualization results of attended regions of images from different prompts. It is also interesting to see how learned inter-class and intra-class prompts are different from each other by using t-sne or PCA.
>
> **A2:** It is a very good suggestion to improve the quality of our paper. We have added a visualization in Section A11 of the appendix, which shows the learned prompts of 10 classes with T-SNE.  Specifically, we randomly select 10 classes from ImageNet and generate the textual embedding with our learned prompts. Then, we obtain $4 \times 10$ embeddings with dimension $d= 1024$. Then we apply the T-SNE to reduce the dimension and visualize the embeddings. As shown in Figure A2, the textual embeddings of the same class with different prompts are clustered well.  Besides, despite being well clustered, we found that the textual embeddings also have intra-diversities.

---

> ### Author Response · Authors · 2022-11-18
> **Response to Reviewer 1oAQ (Part II)**
>
> > **Q3:** It is unclear to see the differences among model variants in Table 2 just from the caption. It will be good if an individual section on introducing the detailed design/training setup of each model in the paper (or in the appendix).
>
> **A3:** We sincerely appreciate this suggestion, which improved our paper's readability. We have added a section in the appendix to provide more details about the different variants in Table 2 (Please refer to Section A3 in the revised appendix), and mentioned it in Table 2 of the main paper.
> Here we briefly describe these methods.
> 1) CoOp: CoOp is the baseline method that only learns a single prompt and matches this single prompt and the global visual feature.
> 2) "G": In this paper, we propose to explore whether we can learn multiple prompts for more comprehensive textual representation and fine-grained visual-textual alignment. "G" denotes that we build multiple prompts (similar to our PLOT) and learn them by matching them with a single global visual feature.
> 3) "G+V": Matching all local prompts to a single visual feature reduces the diversity of the learned prompts. To improve the variety of learned prompts, "G+V" further adds an objective function to increase the distances between every two prompts.
> 4) "G+E": "G+E" is also a method to increase the variety of prompts by separated initializations. It applies predefined different initializations to replace the random initialization, such as "a photo of a", "this is a photo", "this is a", and "one picture of a".
> 5) "M": One key difference between CoOp and our method is to utilize the feature map for more fine-grained information. To evaluate whether our improvement mainly comes from using a feature map, we design a method "M", which removes the OT distance of PLOT and matches local visual features and multiple textual prompts by the average distance of each visual-textual pair.
> 6) "M+V": Similar to "G+V", we add an objective function to increase the distances between every two prompts to the method "M" to increase the variety of prompts.
>
>
> Hope our response can address your concerns.
>
> With best regards,
>
> Authors of submission 898

---

> > ### Comment · Reviewer_1oAQ · 2022-11-22
> > **Thanks for the response**
> >
> > I thank the authors' responses. After reading all the comments and responses, I feel most of the concerns have been addressed. Thus, I increase my rating and remmand for acceptance of this paper.

---

> > ### Author Response · Authors · 2022-11-22
> > **Thank you for recommendation of accepting the paper and increasing the review score**
> >
> > Dear Reviewer 1oAQ,
> >
> > Thank you for your encouraging feedback and recommendation of accepting our paper!
> >
> > Thank you,
> > Authors of submission 898

---

### Official Review · Reviewer_qGjf · 2022-10-28

**Confidence:** 4
**Correctness:** 4
**Technical Novelty And Significance:** 4
**Empirical Novelty And Significance:** 2
**Recommendation:** 6

**Clarity, Quality, Novelty And Reproducibility:**

The paper is of good quality concerning writing and figures. The idea is novel, but the experiments can be further improved as above.

**Strength And Weaknesses:**

Strength:
1) The motivation of enriching prompts for different image regions is practical and well illustrated in the paper. The usage of optimal transport is also novel, which might be inspiring to future works.
2) The ablation study with discussion clearly tackles most my confusion with interesting points.

Weakness:
1) The major concern is the classification performance. As shown in Figure 3, despite the better overall performance, PLOT performs worse than previous prompting methods or linear probe on some of the datasets. Considering the much more complicated training, PLOT is expected to achieve higher accuracy.
2) The authors should provide complete comparisons between PLOT (adapter-based PLOT) and existing adapter-based methods, e.g. Tip-Adapter-F on all 11 datasets, not just ImageNet. As the adapter-based methods are SOTA for CLIP few-shot learning, this can better emphasize the contribution of PLOT.

**Summary Of The Paper:**

The paper proposes PLOT, a prompt learning method for CLIP based on CoOp. PLOT aims to diversify the textual prompts to better depict the category for both its intrinsic and extrinsic contexts. By applying optimal transport, PLOT shows favorable performance over different few-shot benchmarks.

**Summary Of The Review:**

The reviewer expects the authors' response to the experiment weakness.

---

> ### Author Response · Authors · 2022-11-18
> **Response to Reviewer qGjf**
>
> Dear Reviewer qGjf, we are sincerely grateful for your careful reading and valuable suggestions. We have followed your suggestion to improve our paper. Please see the revised paper and appendix for details. Below we give a point-by-point response to the comments.
>
> > **Q1:** The major concern is the classification performance. As shown in Figure 3, despite the better overall performance, PLOT performs worse than previous prompting methods or linear probe on some of the datasets. Considering the much more complicated training, PLOT is expected to achieve higher accuracy.
>
> **A1:** Thank you for your insight. We would like to highlight that we followed most settings of CoOp for a fair comparison and didn't search for special hyper-parameters for PLOT, even if these hyper-parameters may not be optimal for PLOT due to multiple prompts. We can achieve better performance using more appropriate hyper-parameters, e.g., we can obtain 93.6% accuracy on 16-shot Caltech101 if we set the learning rate to 0.001 and epochs to 250, while our performance in Figure 3 is only 92.2.
> Our main goal is to extend current single-prompt learning methods to multi-prompt ones. The improvement of PLOT over CoOp (our baseline method) on most datasets strongly suggests that learning multiple prompts can promote few-shot downstream tasks.
>
> Besides, due to the uncertainty in the experiments, it may be natural that PLOT performs worse in some specific situations.
> Here we provide explanations for three examples.
> First, in Figure 3, PLOT performs comparably with CoOp on StanfordCars. It may be because the discriminative characters in StanfordCars coincide with each other,
> such that one global prompt and one global visual feature can work well. Second, both PLOT and CoOp (the learning method) perform worse than the linear probe on FGVCAircraft. As shown in the discussion in CoOp, it is because that pre-trained CLIP space has been proven powerful, making the linear probe model a strong competitor. Third, PLOT performs worse than CoCoOp on some datasets. It is because CoCoOp learns an extra meta-network to learn the context conditions of each prompt.  For better comparison, we have conducted a new experiment that applies the PLOT to CoCoOp by learning multiple textual prompts instead of the single one (also using the meta-network). Please refer to Section A6 and Table A4 of the appendix for the details. The results on the 11 datasets with 16 shots are also provided below. We found PLOT achieves improvement on most datasets and on average.
>
> | Method |Average |  Caltech  | DTD |  EuroSAT | Food101 | Flowers | FGVC | ImageNet | OxfordPets | StanfordCars  |  SUN397  |  UCF101  |
> |-------|:-------:|:-------:|:-------:|:-------:|:-------:|:-------:|:-------:|:-------:|:-------:|:-------:|:-------:|:-------:|
> | CoCoOp  | 75.7/64.6 |95.0/90.0 | 73.1/50.0 | 88.9/33.5 |  85.0/86.0 | 91.2/67.5 | 25.5/25.7 | 68.3/63.1 | 92.3/94.6 | 61.8/65.3  | 75.1/73.6 | 76.5/61.6 |
> | CoPLOT| 75.9/67.6 | 95.4/90.9 | 72.6/51.4 | 91.0/55.3 |  85.0/85.2 | 89.6/69.2 | 25.6/26.6 | 68.2/63.1 | 92.1/95.9 | 63.2/66.5  |75.2/73.2 | 77.4/66.2 |
>
>
> > **Q2:** The authors should provide complete comparisons between PLOT (adapter-based PLOT) and existing adapter-based methods, e.g. Tip-Adapter-F on all 11 datasets, not just ImageNet. As the adapter-based methods are SOTA for CLIP few-shot learning, this can better emphasize the contribution of PLOT.
>
> **A2:** Thanks a lot for this valuable suggestion! In light of your suggestion, we have added the experimental results of adapter-based PLOT on all 11 datasets in the appendix. Please kindly see Section A5 and Table A3 in the revised appendix for the details. Below, we provide the on-average performance of all 11 datasets in the table. On average, our adapter-based PLOT method consistently outperforms the original Tip-Adapter-F for all settings (shots), which demonstrates that PLOT can also benefit the adapter-based methods. We did not include the other 10 datasets in the original submission since only the hyper-parameters of Tip-Adapter on ImageNet are released when we conducted this experiment.
>
> | Method |  shot-1  | shot-2 |  shot-4 | shot-8 | shot-16 |
> |-------|:-------:|:-------:|:-------:|:-------:|:-------:|
> | Tip-adapter-F  | 64.62|	66.65|	69.67|	72.45|	75.83 |
> | Tip-adapter-F + PLOT| 65.45|	68.63|71.23|73.49|76.20 |
>
> We hope these experiments and explanations can address your concerns. It would be appreciated if furthermore response.
>
> With best regards,
>
> Authors of submission 898

---

> > ### Comment · Reviewer_qGjf · 2022-11-19
> > **Response to Authors**
> >
> > Dear Authors,
> >
> > Thanks for your detailed point-to-point responses, which clearly address my previous concerns. Besides, the 11-dataset comparison of PLOT and Tip-Adapter-F is better to be presented in the main paper as the final version, which can indicate the superiority of PLOT to adapter-based methods.
> >
> > Reviewer qGjf

---

> > > ### Author Response · Authors · 2022-11-19
> > > **Response to Reviewer qGjf**
> > >
> > > Dear Reviewer qGjf,
> > >
> > > Thanks a lot for your quick reply and greatly appreciate your furthermore valuable feedback which helped improve our paper.  We cannot modify the manuscript now due to the rules but will move Table A3 into the main paper in the final version.
> > >
> > > Thanks again for your help on our paper.
> > >
> > > With best regards,
> > >
> > > Authors of submission 898

---

> > ### Author Response · Authors · 2022-11-22
> > **Could you please consider updating your recommendation, given that your previous concerns were clearly addressed?**
> >
> > Dear Reviewer qGjf,
> >
> > We understand you are busy and appreciate your time. Thank you very much for letting us know that your previous concerns have been clearly addressed. In this case, could you please consider updating your recommendation to reflect it (the original review score is 6)?
> >
> > We highly appreciate your further suggestion on the placement of Table A3 and will update the presentation in the final paper as soon as we can.
> >
> > With best regards,
> > Authors of submission 898

---

### Official Review · Reviewer_FMp9 · 2022-10-29

**Confidence:** 5
**Clarity, Quality, Novelty And Reproducibility:** The quality, clarity and originality …
**Correctness:** 4
**Technical Novelty And Significance:** 4
**Empirical Novelty And Significance:** 4
**Recommendation:** 8

**Strength And Weaknesses:**

Strengths:

\+ The problem of adapting CLIP under few-shot setting is recent. Compared to the baseline method CoOp, the improvement of the proposed method is significant.

\+ The motivation that (1) single prompt is insufficient to represent a class, (2) directly learning multiple prompts reduce to one single point, are straightforward and make sense. The proposed optimal transport based method is consistent with the motivation.

\+ The explanation of the comparison between optimal transport and other distances is clear, which supports the motivation.

\+ The ablation studies and analysis in Section 4.4 is well organized and clearly written. It is easy to follow the analysis and figure our the contribution of each component. Also, Figure 2 is well designed and clear to illustrate the pipeline.

\+ The experimental analysis is comprehensive. The analysis on computation time and inference speed is also provided.

Weakness:

(As I reviewed this paper before and all my previously concerns are addressed, I do not have much comments about the weakness.)

This work is built on top of CoOp. It would be interesting to know whether this proposed method can be extend to other prompt-learning-based methods, especially CoCoOp. If not, it would be great to discuss the reasons. Somehow, the method is only applied to CoOp, which is a limitation of this work.

**Summary Of The Paper:**

This paper focused on adapting large-scaled pre-trained vision-language model (i.e., CLIP) to downstream datasets under the few-shot setting. Compared to the recent related work CoOp, this work (1) learns multiple prompts, (2) uses both local features and global features, and (3) measures the similarity of prompts and visual features using optimal transport. Experiments on 11 benchmark visual recognition datasets show that the proposed method outperforms the baseline CoOp. Ablation studies further more analysis on the contribution of each component.

**Summary Of The Review:**

Overall, considering both strengths (simple but effective pipeline, comprehensive experimental analysis) and weakness (generalizability), my initial rating is accept.

Reasons to accept:

\+ The motivation for this work is clear. Single prompts may lose the details of local attributes or contexts.

\+ The idea of applying optimal transport to prompt learning makes sense and meets well with the motivation.

\+ The experiments and analysis are comprehensive. The improvement is significant.

Reasons to reject:

\- This work is only applied to CoOp. It is not clear whether the method can be extended to other prompt-learning-based methods.

---

> ### Author Response · Authors · 2022-11-18
> **Response to Reviewer FMp9**
>
> Dear Reviewer FMp9, we greatly appreciate your time dedicated to reviewing this paper and your kind approval.
> We provide the point-to-point response to your comments below and have updated the paper and appendix accordingly.
>
> > **Q1:** This work is built on top of CoOp. It would be interesting to know whether this proposed method can be extended to other prompt-learning-based methods, especially CoCoOp. If not, it would be great to discuss the reasons. Somehow, the method is only applied to CoOp, which is a limitation of this work.
>
> **A1:** Thanks for your valuable suggestion that helped improve our work. In light of your suggestion, we have applied our method on CoCoOp, and added an experimental evaluation in Section A6 of the appendix. We applied our PLOT to CoCoOp by learning multiple textual prompts (e.g. N=4) instead of the single prompt in CoCoOp, and call it CoPLOT. Specially, we learned multiple prompts and used the same condition (the output of the meta-network) for all local prompts. Then we employed the Optimal Transport to calculate the distance between multiple local prompts and local visual features. We implement both CoCoOp and CoPLOT using the same RN50 backbone and evaluated their performance in the setting of "base-to-new". The results on the 11 datasets with 16 shots are provided in Table A4 of the appendix and shown in the table below. We observed that PLOT achieves improvement on most datasets and on average, which demonstrates that it can be applied to different prompt-learning-based methods.
>
> | Method |Average |  Caltech  | DTD |  EuroSAT | Food101 | Flowers | FGVC | ImageNet | OxfordPets | StanfordCars  |  SUN397  |  UCF101  |
> |-------|:-------:|:-------:|:-------:|:-------:|:-------:|:-------:|:-------:|:-------:|:-------:|:-------:|:-------:|:-------:|
> | CoCoOp  | 75.7/64.6 |95.0/90.0 | 73.1/50.0 | 88.9/33.5 |  85.0/86.0 | 91.2/67.5 | 25.5/25.7 | 68.3/63.1 | 92.3/94.6 | 61.8/65.3  | 75.1/73.6 | 76.5/61.6 |
> | CoPLOT| 75.9/67.6 | 95.4/90.9 | 72.6/51.4 | 91.0/55.3 |  85.0/85.2 | 89.6/69.2 | 25.6/26.6 | 68.2/63.1 | 92.1/95.9 | 63.2/66.5  |75.2/73.2 | 77.4/66.2 |
>
>
> With best regards,
>
> Authors of submission 898

---

### Decision · Program_Chairs · 2023-01-20

**Decision:**

Accept: notable-top-25%

**Justification For Why Not Higher Score:**

While the paper is easy to follow, the write up could still be improved as pointed out by one of the reviewers. In addition, it would be important to see the experiments with PDL in all datasets.

**Justification For Why Not Lower Score:**

All reviewers agree the proposed method is novel, clearly motivated, and outperforms strong baselines.

**Metareview: Summary, Strengths And Weaknesses:**

The paper introduces a novel method for prompt learning in vision-language models. The idea is to create a diverse set of prompts (instead of a single prompt as used in previous methods) to better model local attributes or context, while applying optimal transport to match the vision and text modalities. All reviewers recommend acceptance and the AC agrees with this decision. The motivation of this work is clear and the experimental analysis is comprehensive. Initially, the reviewers raised concerns about the generality of the approach and requested comparisons with additional methods such as CoCoOp. The author response was well received and included additional experiments showing the effectiveness of the proposed method.

**Note From Pc:**

if the above contains the word "oral" or "spotlight" please see: "oral" presentation means -> notable-top-5% and "spotlight" means -> notable-top-25%. As stated in our emails, we are disassociating presentation type from AC recommendations

**Summary Of Ac-Reviewer Meeting:**

N/A